# Functional MRI Time Series Generation via Wavelet-Based Image Transform and Spectral Flow Matching for Brain Disorder Identification

**Hwa Hui Tew**[1][*]**, Junn Yong Loo**[1][*]**, Fang Yu Leong**[1]**, Julia K. Lau**[1]**, Ding Fan**[1]**,
Hernando Ombao**[3]**, Raphaël C.-W. Phan**[1]**, Chee Pin Tan**[2]**, Chee-Ming Ting**[1][†]
[1]School of Information Technology, Monash University Malaysia
[2]School of Engineering, Monash University Malaysia
[3]Statistics Program, King Abdullah University of Science and Technology
{hwa.tew,loo.junnyong,ting.cheeming}@monash.edu

## ABSTRACT

Functional Magnetic Resonance Imaging (fMRI) provides non-invasive access to dynamic brain activity by measuring blood oxygen level-dependent (BOLD) signals over time. However, the resource-intensive nature of fMRI acquisition limits the availability of high-fidelity samples required for data-driven brain analysis models. While modern generative models can synthesize fMRI data, they often remain challenging in replicating their inherent non-stationarity, intricate spatiotemporal dynamics, and physiological variations of raw BOLD signals. To address these challenges, we propose Dual-Spectral Flow Matching (DSFM), a novel fMRI generative framework that cascades dual frequency representation of BOLD signals with spectral flow matching. Specifically, our framework first converts BOLD signals into a wavelet decomposition map via a discrete wavelet transform (DWT) to capture globalized transient and multi-scale variations, and projects into the discrete cosine transform (DCT) space across brain regions and time to exploit localized energy compaction of low-frequency dominant BOLD coefficients. Subsequently, a spectral flow matching model is trained to generate class-conditioned cosine-frequency representation. The generated samples are reconstructed through inverse DCT and inverse DWT operations to recover physiologically plausible time-domain BOLD signals. This dual-transform approach imposes structured frequency priors and preserves key physiological brain dynamics. Ultimately, we demonstrate the efficacy of our approach through improved downstream fMRI-based brain network classification. The code is available at https://github.com/htew0001/DSFM.git.

## 1 INTRODUCTION

Recent advances in deep generative modeling have shown promising capability in synthesizing realistic yet diverse variations of neuroimaging modalities (Yap et al., 2024). Among available modalities, functional MRI (fMRI) signals offer a non-invasive view of neuronal activity, critical for diagnosing neuropsychiatric and neurodevelopmental disorders (Noman et al., 2022; 2024). However, fMRI data collection is costly and yields limited, often imbalanced datasets (Tew et al., 2025b). These shortcomings limit the generalizability of data-driven brain analysis models, ultimately affecting the reliability of computer-aided clinical tools for neurological and psychiatric conditions (Bollmann & Barth, 2021; Ting et al., 2022). To address these challenges, generative models have been explored for fMRI signal synthesis to support data augmentation and downstream applications (Power et al., 2014; Tew et al., 2025b).

---

[*]Equal contribution.
[†]Corresponding author.

Most existing approaches generate brain connectivity directly in the functional connectivity (FC) space, where BOLD signal dependencies are summarized by a single correlation matrix (Biswal & Uddin, 2025). For instance, Tan et al. (2024b) proposes a DCGAN that preserves connectomic structure and improves the performance of downstream FC classifiers. Similarly, BrainFC-CGAN jointly trains adversarial and supervised loss components to preserve the subject identity of real FC on synthetic samples (Tan et al., 2024a). However, such FC representations encode static pairwise relations into dyads and do not effectively capture transient network states within human brain networks (Shabestari et al., 2025).

Recent works have revisited time-domain modeling of fMRI as an alternative to correlation-based functional connectivity (FC). Yuan & Qiao (2024) designs diffusion-TS, a denoising diffusion probabilistic model (DDPM) for fMRI time series data generation, showing improved robustness over GANs and (Variational Autoencoder) VAE-based generative models. Hu et al. (2024) proposes FM-TS that accelerates the sampling step yet provides quality synthetic samples via a flow matching framework. While these methods shift focus from traditional FC to time-series data generation, their feasibility and effectiveness for neuroimaging tasks remain largely unexplored. We argue that limiting generative modeling to FC matrices or the raw time series is inadequate to faithfully reproduce the brain's transient state, multiscale oscillations, and cross-frequency interactions due to difficulties in disentangling physiologically driven fluctuations (e.g., cardiac pulsation, respiratory cycles, motion-induced artifacts) (Biswal & Uddin, 2025). In contrast, a time-frequency/scale representation that captures time and spectral BOLD information can fully reproduce the rich spatiotemporal dynamics of BOLD signals. Motivated by T2I-Diff and ImagenTime, both of which frame time-series signals as an image-generation task (Tew et al., 2025b; Naiman et al., 2024). T2I-Diff specifically remodeled and validated the feasibility of this time-frequency image-based approach for generating BOLD signals. Crucially, the performance gains were modest due to the fixed-resolution Short-Time Fourier Transform (STFT) representation, which neglects fine-grained transients and attenuates frequency amplitude modulations, leading to artifacts during the image-to-signal reconstruction (Tew et al., 2025b).

To address these issues, in this paper, we propose Dual-Spectral Flow Matching (DSFM), an fMRI generation framework that cascades two spectral transformations of BOLD signals and integrates a spectral flow matching for generative modeling. Our framework first decomposed BOLD signals using the discrete wavelet transform (DWT) to form multiresolution time-scale scalogram images. Subsequently, we compute a discrete cosine transform (DCT) that exploits low-frequency BOLD coefficients. These transforms produce a dual-spectral view in which local and global dynamics are jointly represented. Additionally, our framework introduces a spectral-domain flow matching for efficient and high-fidelity generation of the time-scale fMRI scalograms conditioned on subject classes. The generated time-frequency scalograms are then reverted to BOLD signals via image-to-time series transforms. Our main contributions are summarized as follows:

1. Our proposed DSFM framework is the first to jointly leverage DWT and DCT, forming a unified dual-spectral image transform to capture both global and local spatiotemporal and spectral features for fMRI BOLD signal generation and brain disorder classification.

2. We develop a spectral flow matching to model a heat dissipation process in the DCT domain to achieve efficient, coarse-to-fine generation aligned with the frequency hierarchy of the dual-spectral representation. This enables DSFM to leverage the spectral sparsity inherent in fMRI signals to effectively capture diverse brain profiles.

3. Our results show that DSFM demonstrates strong performance on unconditional and conditional spectral image synthesis, and achieves improvement in brain disorder classification compared to recent time-series and fMRI generation baselines.

## 2 METHODS

### 2.1 DISCRETE WAVELET TRANSFORM AND ITS INVERSION

Fig. 1 provides an overview of our proposed framework. Given high-dimensional fMRI signals from $S$ subjects, denoted as $\mathcal{X} = \{x_s\}_{s=1}^{S}$, where each subject $x_s \in \mathbb{R}^{D \times T}$ consists of $D$ regions of interest (ROIs) recorded over $T$ time points, our objective is to learn the underlying real data

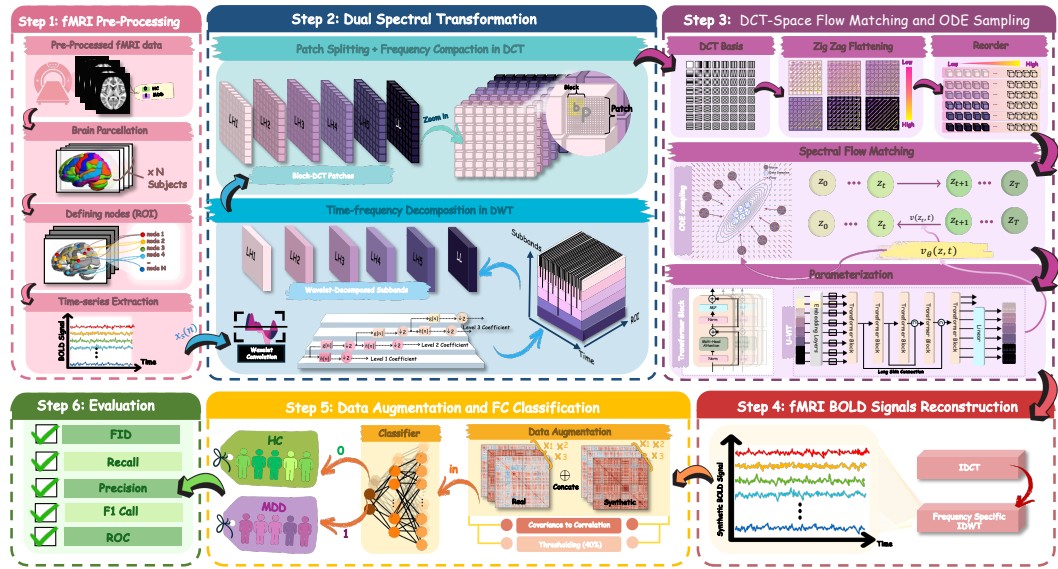

Figure 1: The pipeline of DSFM. ROI-based BOLD time series are first extracted, followed by DWT-based multiresolution decomposition and blockwise 2D DCT for localized spectral encoding. U-ViT is used to model the velocity field in the DCT domain for ODE-based sampling. The reconstructed signals (via IDCT and IDWT) are then used for data augmentation, FC matrix construction, and classification. Finally, fidelity and downstream performance are evaluated.

distribution $p_{\text{data}}(\mathcal{X})$ and generate a synthetic distribution $p_\theta(\mathcal{X})$ that is statistically indistinguishable from the real data. Unlike conventional time-series generative tasks that operate exclusively in the time domain, our approach transforms fMRI time series into time-scale images using the DWT, defined as follows:

$$W(k,j) = \sum_{n=1}^{N} x(n)\,\psi_{j,k}[n], \tag{1}$$

where $x_s(n)$ is the BOLD signal at local time index $n \in \{1, 2, \ldots, N\}$. Here, $\psi_{j,k}[n] = 2^{-k/2}\psi[2^k n - j]$ is the dyadic wavelet basis function, where scale $k \in \{1, 2, \ldots, \lfloor \log_2 N \rfloor\}$ controls the frequency resolution, and translation index $j \in \{1, 2, \ldots, N/2^k\}$ determines the time location, derived from the mother wavelet $\psi_{j,k}[n]$. To construct a wavelet decomposition map, we upsample each wavelet subband to the original time length and stack them along the scale axis, forming a multiresolution wavelet decomposition map. Thus forming a full wavelet coefficient tensor $W(i,j,k) \in \mathbb{R}^{D \times T_\psi \times C}$, where $T_\psi = N/2^C$ and $C = \lfloor \log_2 N \rfloor$, that captures both low-frequency trends and high-frequency transients in the fMRI BOLD signals. We further perform component-wise normalization to accentuate the difference between high and low coefficients over brain regions and time. As shown in Fig. 2, this allows the time-series signals to be represented as multichannel images with preserved spectral-temporal characteristics.

To reconstruct the original signals from the generated scalogram representation, we first denormalize the predicted wavelet components $\hat{W}^{(i)}(j,k) \in \mathbb{R}^{T \times C}$ of each $i^{\text{th}}$ ROI. The coefficients are then downsampled according to their corresponding dyadic scales and computed the inverse DWT (IDWT) to obtain the fMRI BOLD signals as follows:

$$\hat{x}(n) = \frac{1}{N}\sum_{k=1}^{C}\sum_{j=1}^{T} W^{(i)}(j,k)\,\psi_{j,k}[n]. \tag{2}$$

Finally, these wavelet subbands are reconstructed through a hierarchical combination of approximation and detail components across all scales to obtain the reconstructed time-domain signal $\hat{x}_s$ for each subject. This process ensures that the inherited spectral-temporal characteristics of the original fMRI BOLD signals are well-preserved.

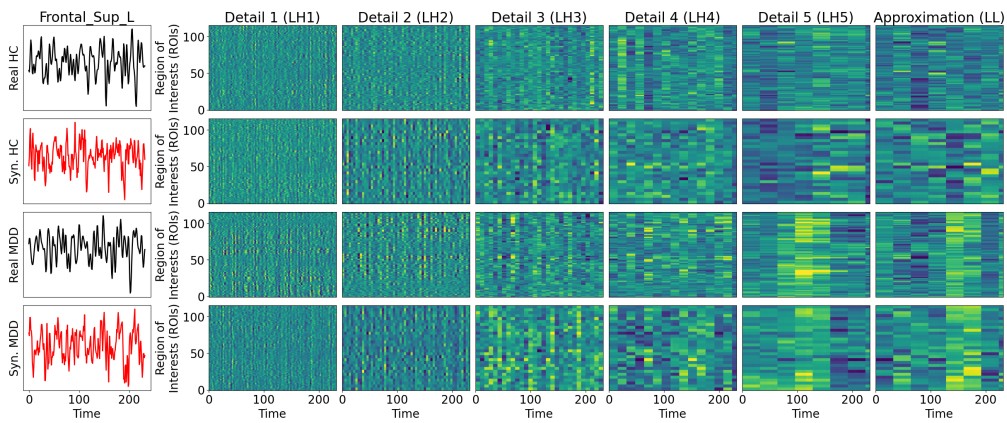

Figure 2: Original (Rows 1&3) vs. synthetic BOLD signals (Rows 2 &4) and generated normalized scalograms. Our framework generates new synthetic BOLD signals as opposed to correlation matrices or functional connectivity, with distributional statistics that closely match the original samples.

## 2.2 DISCRETE COSINE TRANSFORM FOR BOLD SIGNALS

To extract localized energy compactions of low-frequency spontaneous BOLD coefficients. We divide each subband map $\hat{W}^{(k)}(i,j) \in \mathbb{R}^{D \times T_\psi}$ of each $k^{\text{th}}$ wavelet scale into non-overlapping 2D blocks of size $B \times B$, resulting in a set of blocks (patches):

$$W^{(k)} \equiv \left\{ W_p^{(k)}(x,y) \in \mathbb{R}^{B \times B} \right\}_{p=1}^{P}, \tag{3}$$

where $P$ is the number of blocks per subband image and $B$ is the block size. Each block is then transformed via 2D type-II DCT, as follows:

$$D^{(k)}(u,v) = \alpha(u)\,\alpha(v) \sum_{x=1}^{B} \sum_{y=1}^{B} W^{(k)}(x,y) \cos\left[\frac{(2x+1)u\pi}{2B}\right] \cos\left[\frac{(2y+1)v\pi}{2B}\right], \tag{4}$$

where $\alpha(u) = \sqrt{\frac{1}{B}}$ if $u = 0$, and $\alpha(u) = \sqrt{\frac{2}{B}}$ otherwise. The resulting $D^{(k)}(u,v) \in \mathbb{R}^{B \times B}$ at each scale $k$ represents the DCT coefficients within each block.

To recover the full image representation, we apply the inverse 2D DCT (IDCT) to each block $D^{(k)}(u,v)$. The signal block is reconstructed via the following inverse transform:

$$\hat{W}^{(k)}(x,y) = \sum_{u=1}^{B} \sum_{v=1}^{B} \alpha(u)\,\alpha(v) D^{(k)}(u,v) \cos\left[\frac{(2x+1)u\pi}{2B}\right] \cos\left[\frac{(2y+1)v\pi}{2B}\right]. \tag{5}$$

Once all blocks have been transformed back into the original spatial domain, we stitch the patches to recover the full subband map. Since the DCT is applied to non-overlapping blocks, the reconstruction involves simply tiling the inverse-transformed blocks back into their original positions within the subband image. This blockwise DCT preserves localized low-frequency structure in the ROI–time space while discarding high-frequency noise components. The resulting set of filtered subbands can then be passed back into the IDWT to recover the time-domain fMRI BOLD signal $\hat{x}_s(n)$, ensuring global and local spectral characteristics are retained.

## 2.3 SPECTRAL FLOW MATCHING IN DCT DOMAIN

Recent studies have empirically demonstrated that pixel-based diffusion models exhibit approximate autoregressive behavior in the frequency domain (Dieleman, 2024; Falck et al., 2025). Specifically, diffusion models (Ho et al., 2020; Song et al., 2021) tend to eliminate high-frequency components early in the forward process, followed by progressively lower-frequency components as the diffusion timestep advances. While prior studies focus on the Fourier basis, this property also holds in the

DCT domain (Skorokhodov et al., 2025; Ning et al., 2025), which offers practical advantages: real-valued orthogonality, energy compaction in low-frequency bands, and compatibility with block-wise architectures.

Modeling diffusion directly in the frequency domain enables the exploitation of spectral sparsity for designing frequency-aware noise schedules. However, existing frequency-domain generative models (Hoogeboom & Salimans, 2023; Rissanen et al., 2023) remain constrained to the diffusion framework, which relies on stochastic differential equation (SDE) sampling and typically requires hundreds to thousands of steps for high-quality synthesis. In contrast, flow-matching approaches based on ordinary differential equations (ODEs) provide a deterministic alternative with significantly lower sampling complexity. In this work, we introduce a spectral flow-matching framework that extends frequency-based generative modeling beyond the diffusion paradigm.

First, consider a forward-time heat dissipation process (Rissanen et al., 2023) as an alternative to the conventional isotropic diffusion, described by the following stochastic partial differential equation (SPDE):

$$dx_t(c) = \eta(t) \, \Delta_c \, x_t(c) \, dt + G(t) \, dW(t), \tag{6}$$

where $x_t : \mathbb{R}^2 \times \mathbb{R}_+ \to \mathbb{R}$ is an idealized continuous-space representation of a single image channel at time $t \in [0, 1]$, and $\Delta_c = \nabla_c \cdot \nabla_c$ is the Laplace operator with respect to the spatial image coordinates $c$; $\eta(t)$ and $G(t)$ are time-dependent scalar drift and matrix-valued diffusion coefficients, respectively. The corresponding reverse-time probability flow ODE (Song et al., 2021) is given by:

$$\frac{dx_t}{dt} = \eta(t) \, \Delta_c \, x_t(c) - \frac{1}{2} \, G(t)G(t)^T \, \nabla_{x_t} \log p(x_t). \tag{7}$$

Subsequently, define the forward and inverse DCT transforms formally as

$$z_t = V^T x_t = \mathrm{DCT}(x_t), \quad x_t = V z_t = \mathrm{IDCT}(z_t), \tag{8}$$

where $V$ denotes the matrix of orthonormal DCT basis eigenvectors. It then follows that the Laplacian operator in equation 6 can be diagonalized via the eigendecomposition $\Delta_c = V \Lambda V^T$, where $\Lambda$ denotes the diagonal matrix of DCT mode-specific Laplacian eigenvalues. Applying DCT to the forward-time SPDE equation 6 yields

$$dz_t = -\eta(t) \, \Lambda \, z_t \, dt + G(t) \, dW(t), \tag{9}$$

where $W(t)$ is a standard Wiener process, but in the DCT domain. To obtain a frequency-ordered representation, we apply zig-zag flattening, which maps the two-dimensional DCT coefficient grid into a one-dimensional sequence sorted from low (upper-left) to high (bottom-right) frequencies. Subsequently, we apply per–DCT-mode signal-to-noise scaling to ensure that the DCT coefficients conform to the forward perturbation process of the proposed DCT flow governed by the SPDEs in equation 6 and equation 9. Moreover, given that $V$ is orthonormal, the following change-of-variables holds for any differentiable function $f$:

$$V^T \nabla_x f(x) = V^T \left( \frac{\partial z}{\partial x} \right)^T \nabla_z f(z) = \underbrace{V^T V}_{I} \nabla_z f(\underbrace{V^T x}_{z}) = \nabla_z f(z).$$

By letting $f(x_t) = \log p(x_t)$, the score transforms as $V^T \nabla_{x_t} \log p(x_t) = \nabla_{z_t} \log p(z_t)$. Since $\Lambda$ is diagonal and the DCT basis orthogonalizes the frequency modes, applying DCT to equation 7, the reverse-time probability flow ODE admits the following mode-wise decomposition:

$$\frac{dz_t[k]}{dt} = -\eta(t) \, \lambda_k \, z_t[k] - \frac{1}{2} \, g(t, k)^2 \, \nabla_{z_t[k]} \log p(z_t), \tag{10}$$

where $\lambda_k$ is the $k$-th diagonal entry of $\Lambda$, corresponding to the Laplacian eigenvalue of the $k^{\mathrm{th}}$ DCT basis component, which evolves independently under the ODE dynamics.

The following proposition bridges between this DCT mode-wise probability flow ODE and the conditional velocity (vector) field in flow matching (Lipman et al., 2023).

**Proposition 1.** *A mode-wise conditional perturbation kernel (isotropic in each DCT mode) is*

$$p(z_t[k] \,|\, z_0[k]) = \mathcal{N}\big(\mu(t, k) \, z_0[k], \sigma(t, k)^2\big), \tag{11}$$

Table 1: Comparison of unconditional Netsim dataset generation across SOTA and proposed model.

|  | CoT-GAN | DiffTime | DiffWave | TimeVAE | TimeGAN | Diffusion-TS | T2I-Diff | DSFM (Ours) |
|---|---|---|---|---|---|---|---|---|
| cFID $\downarrow$ | 7.813±.550 | 0.340±.015 | 0.244±.018 | 14.449±.969 | 0.126±.002 | 0.105±.006 | 1.384±.107 | 0.193±.017 |
| Corr. $\downarrow$ | 26.824±.449 | 1.501±.048 | 3.927±.049 | 17.296±.526 | 23.502±.039 | 1.411±.042 | 4.121±.094 | 4.552±.041 |
| Disc. $\downarrow$ | 0.492±.018 | 0.245±.051 | 0.402±.029 | 0.476±.044 | 0.484±.042 | 0.167±.023 | 0.400±.059 | 0.497±.001 |
| Pred. $\downarrow$ | 0.185±.003 | 0.100±.000 | 0.101±.000 | 0.113±.003 | 0.126±.002 | 0.099±.000 | 0.102±.001 | 0.104±.000 |

*with mean and standard deviation (std) schedules*

$$\mu(t,k) = \alpha(t)\,\omega(t,k), \quad \omega(t,k) = e^{-\lambda_k \tau(t)}, \quad \sigma(t,k)^2 = 1 - \mu(t,k)^2, \tag{12}$$

*where $\alpha(t)$ is the mean schedule of a variance-preserving (VP) diffusion process and $\tau(t) = \int_0^t \eta(s)\,ds$, satisfies the heat dissipation process (6) and (9). The mode-wise diffusion coefficients are then given by*

$$g(t,k)^2 = 2\,\sigma(t,k)\left(\dot{\sigma}(t,k) - f(t,k)\,\sigma(t,k)\right), \tag{13}$$

*where $f(t,k) = \frac{\dot{\alpha}(t)}{\alpha(t)} - \eta(t)\,\lambda_k$, and $\dot{\mu}(t,k)$, $\dot{\sigma}(t,k)$ denote time-derivatives of the mean and std schedules in (12).*

*Proof.* Refer to the Supplementary Material. □

**Proposition 2.** *A mode-wise conditional velocity field*

$$\frac{dz_t[k]}{dt}\bigg|_{z_0[k]} = v(z_t|z_0; t, k) = \dot{\mu}(t,k)\,z_0[k] + \dot{\sigma}(t,k)\,\epsilon, \tag{14}$$

*where $\epsilon \sim \mathcal{N}(0,1)$, is equivalent to the conditional probability flow ODE*

$$\frac{dz_t[k]}{dt}\bigg|_{z_0[k]} = -\eta(t)\,\lambda_k\,z_t[k] + \frac{1}{2}\,g(t)^2\,\nabla_{z_t[k]} \log p(z_t|z_0). \tag{15}$$

*Furthermore, it follows that the marginal velocity field*

$$\frac{dz_t[k]}{dt} = v(z_t; t, k) = \mathbb{E}_{p_{data}(z_0|z_t)}\left[v(z_t|z_0; t, k)\,|\,z_t\right], \tag{16}$$

*given by the law of the unconscious statistician (Lipman et al., 2024), satisfies the marginal mode-wise probability flow ODE (10).*

*Proof.* Refer to the Supplementary Material. □

Given this correspondence between the probability flow ODE from diffusion models and flow matching, we parameterize the velocity field $v_\theta$ using a deep neural network (U-ViT (Bao et al., 2023)) and train it via the following conditional spectral flow matching (CSFM) loss:

$$\mathcal{L}^{\text{CSFM}}(\theta) = \mathbb{E}_{t, p(z_t\,|\,z_0) p_{\text{data}}(z_0)} \big\| v_\theta(z_t; t, k) - v(z_t|z_0; t, k) \big\|^2, \tag{17}$$

where $v(z_t|z_0; t, k)$ is the conditional velocity field in (14), with $z_t$ sampled from the per-mode conditional perturbation kernel (11), and $t \sim \mathcal{U}(0,1)$ is uniformly sampled. Notably, this CSFM loss recovers the standard flow matching loss under the OT-CFM schedules $\mu(t) = 1 - t$ and $\sigma(t) = t$, where the time convention adopted here is the reverse of that in (Lipman et al., 2023). Hence, our framework generalizes flow matching to a heat dissipation process in the DCT domain. In our experiments, we use $\alpha(t)$ from the variance-preserving (VP) cosine schedule and set $\tau(t) = \sigma_{\max} \sin^2\left(\frac{\pi}{2}t\right)$ following (Hoogeboom & Salimans, 2023), which observes optimal performance with $\sigma_{\max} = 20$.

To enable class-conditioned generation, we employ classifier-free guidance (Ho & Salimans, 2021) by conditioning the velocity model on the class label $c$, i.e., $v_\theta(z_t; t, k, c)$ and set $c = \varnothing$ for the unconditional model. The conditional and unconditional models are trained jointly by randomly replacing the class label $c$ with the null token $\varnothing$ with probability $p_\varnothing$. During sampling, the classifier-free guided velocity is obtained as a weighted combination of the model outputs (Zheng et al., 2023). Finally, DCT samples are generated by numerically integrating the learned flow velocity using an adaptive ODE solver.

Table 2: The best generation quality and classification performance on the MDD dataset for different generative models using the AAL atlas parcellation, trained on ground-truth data augmented at three levels. "–" denotes FC-based generation. *Full results* refer to the Supplementary Material.

| Metric | W/O Aug.
Real (R.) | Vanilla-GAN
R. + Synth. 3× | 2D-DCGAN
R. + Synth. 3× | TimeGAN
R. + Synth. 1× | Diffusion TS
R. + Synth. 1× | T2I-Diff
R. + Synth. 1× | DSFM (Ours)
R. + Synth. 1× |
|---|---|---|---|---|---|---|---|
| Context-FID ↓ | – | – | – | $4.98 \pm 0.65$ | $2.06 \pm 0.21$ | $7.45 \pm 0.42$ | $1.51 \pm 0.41$ |
| Correlational ↓ | – | – | – | $197.05 \pm 17.75$ | $64.16 \pm 3.92$ | $62.32 \pm 1.04$ | $57.30 \pm 2.89$ |
| Accuracy ↑ | $58.90 \pm 2.98$ | $58.86 \pm 2.24$ | $62.88 \pm 4.99$ | $66.78 \pm 1.66$ | $67.29 \pm 1.81$ | $66.87 \pm 3.22$ | $70.84 \pm 5.89$ |
| Recall ↑ | $58.90 \pm 2.98$ | $58.86 \pm 2.24$ | $62.88 \pm 4.99$ | $66.78 \pm 1.66$ | $67.29 \pm 1.81$ | $66.87 \pm 3.22$ | $70.84 \pm 5.89$ |
| Precision ↑ | $59.56 \pm 2.74$ | $59.91 \pm 2.57$ | $63.12 \pm 5.02$ | $67.14 \pm 1.70$ | $67.55 \pm 1.97$ | $67.06 \pm 3.34$ | $70.99 \pm 5.80$ |
| F1-Score ↑ | $58.39 \pm 3.09$ | $57.64 \pm 1.96$ | $62.48 \pm 5.25$ | $66.48 \pm 1.96$ | $67.21 \pm 1.87$ | $66.83 \pm 3.21$ | $70.77 \pm 5.97$ |
| ROC ↑ | $59.00 \pm 2.56$ | $58.57 \pm 2.05$ | $62.67 \pm 5.15$ | $67.26 \pm 2.81$ | $64.57 \pm 2.76$ | $67.26 \pm 6.00$ | $71.49 \pm 5.73$ |

Table 3: The best generation quality and classification performance on the ABIDE dataset for different generative models using the Schaefer parcellation, trained on ground-truth data augmented at three levels. "–" denotes FC-based generation. *Full results* refer to the Supplementary Material.

| Metric | W/O Aug.
Real (R.) | Vanilla-GAN
R. + Synth. 2× | 2D-DCGAN
R. + Synth. 1× | TimeGAN
R. + Synth. 1× | Diffusion TS
R. + Synth. 1× | T2I-Diff
R. + Synth. 1× | DSFM (Ours)
R. + Synth. 1× |
|---|---|---|---|---|---|---|---|
| Context-FID ↓ | – | – | – | $4.25 \pm 1.30$ | $0.51 \pm 0.07$ | $0.82 \pm 0.21$ | $0.07 \pm 0.01$ |
| Correlational ↓ | – | – | – | $238.70 \pm 5.49$ | $26.13 \pm 3.83$ | $25.75 \pm 1.67$ | $13.42 \pm 2.17$ |
| Accuracy ↑ | $64.67 \pm 1.56$ | $64.87 \pm 1.05$ | $65.63 \pm 1.91$ | $66.59 \pm 2.50$ | $66.60 \pm 1.41$ | $69.69 \pm 1.55$ | $71.54 \pm 1.87$ |
| Recall ↑ | $64.67 \pm 1.56$ | $64.87 \pm 1.05$ | $65.63 \pm 1.91$ | $66.59 \pm 2.50$ | $66.60 \pm 1.41$ | $69.69 \pm 1.55$ | $71.54 \pm 1.87$ |
| Precision ↑ | $65.77 \pm 2.77$ | $65.06 \pm 1.28$ | $66.63 \pm 2.44$ | $67.93 \pm 1.87$ | $66.60 \pm 1.41$ | $69.78 \pm 1.74$ | $73.08 \pm 3.00$ |
| F1-Score ↑ | $64.12 \pm 1.39$ | $64.75 \pm 0.93$ | $65.26 \pm 1.90$ | $65.92 \pm 2.92$ | $66.58 \pm 1.41$ | $69.65 \pm 1.53$ | $70.98 \pm 2.30$ |
| ROC ↑ | $67.28 \pm 4.15$ | $68.07 \pm 2.88$ | $68.44 \pm 1.23$ | $67.49 \pm 3.26$ | $68.85 \pm 3.64$ | $71.88 \pm 2.47$ | $73.78 \pm 3.64$ |

## 3 EXPERIMENT

### 3.1 SETTINGS

**Data Acquisition and Pre-processing.** In our experiments, we evaluated the proposed method on one simulated and two real-world brain disorder datasets. (1) Major Depressive Disorder (MDD): We preprocessed the resting-state fMRI (rs-fMRI) dataset from the REST-meta-MDD Consortium database (Yan et al., 2019) using the Data Processing Assistant for Resting-State fMRI (DPARSF) (Yan & Zang, 2010). This dataset comprises 250 Healthy Controls (HC) subjects and 227 individuals diagnosed with Major Depressive Disorder (MDD). All scans were acquired using a Siemens Tim Trio 3T scanner TR/TE = 2000/30 ms, and a slice thickness of 3mm. The brain was parcellated into 116 ROIs, covering cortical and subcortical areas, and the mean BOLD signal for each ROI was extracted across 232 time points using the Automated Anatomical Labeling (AAL) atlas. (2) Autism Brain Imaging Data Exchange (ABIDE): We preprocessed rs-fMRI scans from multiple international sites. This dataset includes 488 Autism Spectrum Disorder (ASD) patients and 537 normal controls (NC) from the ABIDE database. The brain was parcellated into 100 ROIs using the Schaefer atlas, We then extracted mean BOLD signal for each ROI over 200 time points (Di Martino et al., 2014). (3) NetSim: We used the NetSim dataset, a simulated benchmark for evaluating causal discovery in neuroimaging. NetSim provides biologically realistic simulations of BOLD time series, we selected Simulation 4 with 50 channels from the original dataset (Smith et al., 2011).

### 3.2 IMPLEMENTATION DETAILS

**DSFM Training.** The proposed DSFM framework generates fMRI signals corresponding to the subjects' condition. The classifiers then discriminate between control and clinical groups for each dataset. We train the DSFM using an AdamW optimizer with a learning rate of $2e^{-4}$ over 300k iterations. All experiments employ a Haar wavelet basis with a 5-level decomposition, yielding real-valued images of size $116 \times 232$ (MDD) and $100 \times 200$ (ABIDE). We compare numbers of function evaluations (NFE) of 20, 50, and 100 steps.
**Connectivity Network Construction.** The subject-specific functional connectivity is derived using the Ledoit-Wolf (LDW) regularized shrinkage covariance estimator to preserve the strongest $\tau = 40\%$ connections, resulting in a sparse $116 \times 116$ (MDD) and $100 \times 100$ (ABIDE) FCs with all other connections set to zero.

Table 4: Ablation analysis of frequency-specific FC classification by incorporating individual and grouped wavelet subbands on the MDD dataset.

| | Wavelet Subbands | | | | | | Accuracy ↑ | | Precision ↑ | | F1-Score ↑ | | ROC ↑ | |
|---|---|---|---|---|---|---|---|---|---|---|---|---|---|---|
| Setting | LH1 | LH2 | LH3 | LH4 | LH5 | LL | Value | Drop (%) | Value | Drop (%) | Value | Drop (%) | Value | Drop (%) |
| Full-band | ✓ | ✓ | ✓ | ✓ | ✓ | ✓ | 70.84 | – | 70.99 | – | 70.77 | – | 71.49 | – |
| Low-pass | ✗ | ✗ | ✓ | ✓ | ✓ | ✓ | 66.89 | -5.58 | 66.96 | -5.68 | 66.77 | -5.65 | 65.79 | -7.97 |
| Mid-pass | ✓ | ✓ | ✗ | ✗ | ✓ | ✓ | 63.30 | -10.64 | 63.74 | -10.21 | 63.05 | -10.91 | 60.41 | -15.50 |
| High-pass | ✓ | ✓ | ✓ | ✓ | ✗ | ✗ | 65.40 | -7.68 | 65.55 | -7.66 | 65.18 | -7.90 | 63.66 | -11.0 |
| Band-pass 1 | ✗ | ✓ | ✓ | ✓ | ✓ | ✓ | 66.45 | -6.20 | 66.53 | -6.28 | 66.39 | -6.19 | 68.38 | -4.35 |
| Band-pass 2 | ✓ | ✓ | ✓ | ✓ | ✓ | ✗ | 66.66 | -5.90 | 66.88 | -5.79 | 66.60 | -5.89 | 66.74 | -6.64 |
| Band-pass 3 | ✗ | ✓ | ✓ | ✓ | ✓ | ✗ | 66.88 | -5.59 | 66.76 | -5.96 | 67.06 | -5.24 | 67.77 | -5.20 |

**Data Augmentation and Classifier Training.** The trained DSFM is used to augment real fMRI signals by factors of $1\times$, $2\times$, and $3\times$. For our classifier, the L2 regularization weight decay is from $10^{-8}$ to $10^{-2}$, the scheduler learning rate reduction factor is from 0.1 to 0.9, and the batch size is from 5 to 16, the same as in (Tew et al., 2025b). All hyperparameters are selected based on a 5-fold stratified cross-validation.

## 3.3 OVERALL PERFORMANCE

We first trained our proposed DSFM unconditionally to produce comparable outputs in Table 1. Then, we followed the standard setting for the quality and classification evaluation of the conditional time-series generation as described in section D. Our primary goal is to achieve a better cFID score to model complex spatiotemporal patterns and excel in capturing conditional distribution over time.

**Classification Score.** To validate the fidelity of the generated samples, we evaluate the classification performance of BrainNetCNN (Kawahara et al., 2017), comparing DSFM to GAN and diffusion-based baselines on our fMRI dataset. Here, we use the parameter setting of NFE = 100 in subsequent downstream analyses, as supported by the quality metrics of distinguishing HC and MDD subjects in Table 7. Table 2 and 3 reports the classification results on the 5-fold cross-validation test set. Notably, DSFM achieves the highest accuracy under $1 \times$ (MDD) and $1 \times$ (ASD) data augmentation setting. Moreover, our model exhibits lower variance across increased augmentation levels, indicating strong generalization and robustness. These results confirm that DSFM not only enriches sample diversity but also preserves discriminative neurophysiological functional patterns critical for clinical tasks. Figure 3 further demonstrates that our proposed DSFM model excels in generating class-conditioned synthetic data whose statistical distribution closely matches that of the original samples.

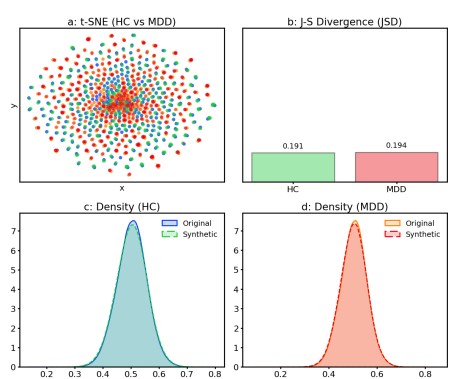

Figure 3: We plot the 2D t-SNE embedding of HC and MDD synthetic data generated by our method (top left). Then, we compare with the distributions using Jensen-Shannon Divergence and probability density functions (top right and bottom).

## 3.4 ABLATION STUDIES

We first conducted ablation study on six wavelet detail bands, i.e., LH1: 0.125 - 0.250Hz, LH2: 0.0625 - 0.125Hz, LH3: 0.03125 - 0.0625Hz, LH4: 0.015625 - 0.03125Hz, LH5: 0.007825 - 0.015625Hz, and a coarse approximation LL: 0 - 0.007825Hz, contrasting each setting with the full 0 to 0.25Hz spectrum. Table 4 assesses the impact of different wavelet subbands on model performance. The steepest decline occurred when the mid-frequency LH3–LH4 pair was removed, highlighting the pivotal role of 0.01–0.06 Hz oscillations to capture disease-specific interactions due to insufficient contextual information. Suppressing either the highest (LH1–LH2) or the very lowest components (LL and LH5) produced a comparable, still significant degradation (5–8%), indicating that both rapid fluctuations and slow drifts provide complementary cues. Conversely, removing in-

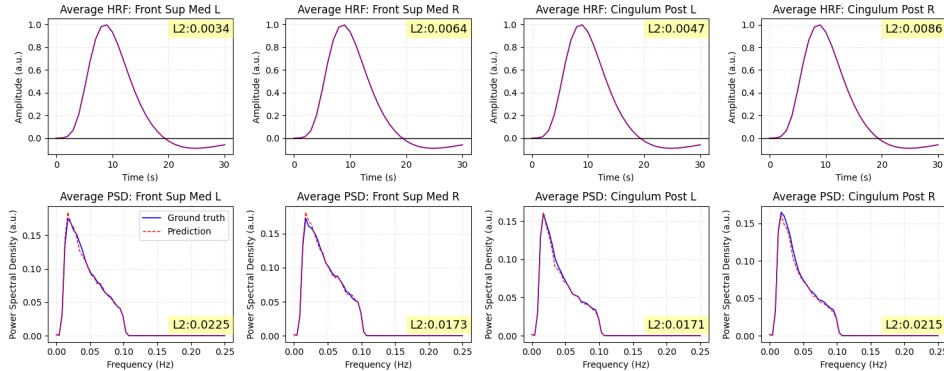

Figure 4: Visualization of the average resting-state hemodynamic response function (rsHRF) and power spectral density (PSD) of real and synthetic BOLD in the Medial Prefrontal Cortex (mPFC) and Posterior Cingulate Cortex (PCC) region of the Default Mode Network (DMN). Highlighted L2 norm quantifies the generation and synthetic results closely resemble the real physiological profiles.

dividual bands such as LH1 and LL also reduced performance by 5–7%, indicating that long-range and slow drifts carry global synchrony patterns essential for classification. Interestingly, we observe that although BOLD fluctuations predominantly lie in the low-frequency band, removing any sub-bands impaired performance, indicating that disease-related features are distributed across the entire frequency spectrum.

Table 5 presents ablation analyses of different configurations evaluating normalization strategies, block sizes and wavelet bases influence the generative quality of our dual-spectral representation. In particular, 1) and 2) show a comparison of MinMax normalization (MM) with the Entropy-Consistent Scaling (ECS). Notably, MinMax scales each wavelet coefficient independently, broadening the distribution of high-frequency coefficients, which results in slower training and reduced performance. In contrast, ECS preserves the global spectral coefficient by normalizing DCT frequency components using a percentile-trimmed bound derived from the lowest frequency component, providing better cFID and correlation scores by maintaining the original coefficient distribution.

Table 5: Ablation studies of block sizes, wavelet bases, normalization strategies, different spectral representations and generative models. *Full results* refer to the Supplementary Material.

| Configurations | cFID↓ | Corr↓ |
|---|---|---|
| 1) $B = 4$, MM | 1.505±0.41 | 57.3±2.89 |
| 2) $B = 4$, ECS | 0.098±0.01 | 18.2±1.41 |
| 3) $B = 2$, Haar | 0.121±0.03 | 19.7±3.03 |
| 4) $B = 4$, Haar | 0.098±0.01 | 18.2±1.41 |
| 5) $B = 4$, dB-4 | 0.199±0.10 | 20.7±3.25 |
| DSFM (Ours) | 0.098±0.01 | 18.2±1.41 |

Experiments with smaller and larger block size $B$ in 3) and 4) achieve comparable generation performance, with a tradeoff of a smaller $B$ will lead to slower training, larger $B$ leads to the loss of fine-grained local dependencies. Finally, the ablations in 4) and 5) using different mother wavelets produces similar generation results on the MDD dataset. This further exemplifies that the underlying fMRI signals do not exhibit strong wavelet-specific bases sensitivity. Additional ablations in Section E.2 further illustrate results for different spectral representations (Fourier and wavelet transforms) and types of generative models (flow matching and diffusion models), against our proposed DSFM.

Table 6: Similarity between synthetic and real FC networks across FC edges, node strength, and edge betweenness centrality. Higher values indicate better preservation of real FC topology.

| Metric | Vanilla-GAN | 1D-DCGAN | 2D-DCGAN | WGAN | WGAN-GP | DSFM (Ours) |
|---|---|---|---|---|---|---|
| FC Edges ↑ | 0.53 ± 0.06 | 0.10 ± 0.11 | 0.54 ± 0.49 | 0.51 ± 0.47 | 0.52 ± 0.17 | 0.99 ± 0.00 |
| Node Strength ↑ | 0.67 ± 0.08 | 0.30 ± 0.16 | 0.53 ± 0.08 | 0.64 ± 0.09 | 0.62 ± 0.03 | 0.99 ± 0.00 |
| Edge Betweenness Centrality ↑ | 0.11 ± 0.02 | 0.06 ± 0.02 | 0.14 ± 0.02 | 0.14 ± 0.02 | 0.15 ± 0.02 | 0.77 ± 0.09 |

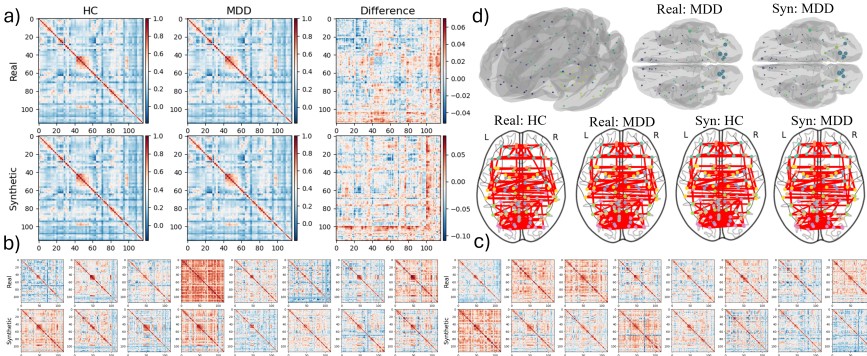

Figure 5: a) Group-averaged connectivity patterns of real and synthetic HC/MDD connectivity patterns and their differences. b) and c) Subject-level connectivity patterns of real and synthetic data from HC and MDD, respectively. d) 3D cortical surface and brain networks visualizations showing node strength (top) and network organization (bottom) for both real and synthetic HC/MDD data.

## 4 NEUROPHYSIOLOGICAL PLAUSIBILITY ANALYSIS

Figure 4 presents the qualitative and quantitative comparisons of resting-state hemodynamic response function (rsHRF) and power spectral density (PSD) between real and synthetic signals in two key hubs of the Default Mode Network (DMN). The near-perfect overlap of the HRF plots indicates that DSFM preserves the canonical temporal dynamics of the hemodynamic process, rather than merely matching marginal statistics. Likewise, the close alignment of the PSD curves indicates that the synthetic samples exhibit realistic fMRI spectral characteristics, accurately capturing the dominant low-frequency peaks and the spectral decay across low- and high-frequency components. The low L2 error for both HRF and PSD provides evidence that DSFM effectively learns underlying spectral-temporal dynamics of the BOLD signals. Overall, these analyses suggest that our model generates neurophysiologically plausible synthetic signals that are suitable for different downstream tasks, as further supported by the brain disorder classification performance in Tables 2 and 3.

## 5 FUNCTIONAL CONNECTIVITY (FC) ANALYSIS AND VISUALIZATION

Table 6 further evaluates the fidelity of the generated data FC matrices derived from real and synthetic fMRI BOLD signals. Across all graph similarity metrics, DSFM shows higher Pearson correlation with the real data than other GAN-based models, indicating more realistic synthesis of FC networks in both connectivity edges and network topology. These results demonstrate that DSFM not only reproduces plausible spectral-temporal BOLD signals dynamics but also faithfully captures higher-order network transformations, reflecting more coherent interdependencies among FC edges than existing GAN-based generative models. Figure 5 visualizes group-averaged connectivity, thresholded at 0.6 to highlight significant edge connections. Our analysis reveals that the synthetic FC closely aligns with the functional changes observed in the real FC distribution. Furthermore, the HC and MDD connectograms between both real and synthetic FC indicate a reduction in intra-network connectivity within the left superior frontal gyrus (FrontalSupL) and weakened coupling between the left middle frontal gyrus (FrontalMidL) and the left anterior cingulate cortex (CingulumAntL). The results suggest impaired cognitive functions associated with difficulties in decision-making and emotion regulation, indicating the biological plausibility of the generated data.

## 6 CONCLUSIONS AND FUTURE WORK

In this paper, we propose DSFM, which effectively captures both temporal dynamics and spectral evolution underlying the ground-truth data distribution for accurate brain signals generation. Future work will further validate MDD and ASD classification using graph-based deep learning models (Tew et al., 2024; 2025a) and incorporate energy-based models to identify out-of-distribution (OOD) patterns in brain spectrograms/scalograms associated with neurological disorders (Loo et al., 2025).

ACKNOWLEDGMENTS

This work was supported in part by the Monash University Malaysia and the Ministry of Higher Education, Malaysia under Fundamental Research Grant Scheme FRGS/1/2023/ICT0 2/MUSM/02/1, and by the King Abdullah University of Science and Technology (KAUST) Grant. The authors also acknowledge the support of the Advanced Computing Platform (ACP), Monash University Malaysia, for providing computational resources.

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

# A    APPENDIX

This appendix provides self-contained additional material for the submission titled *"Functional MRI Time Series Generation via Wavelet-Based Image Transform and Spectral Flow Matching for Brain Disorder Identification"*. It includes a detailed about related works, proofs and derivations, the evaluation metrics, full experimental results, limitations, reproducibility statement, as well as the use of large language models (LLMs).

# B    RELATED WORKS

## B.1    GENERATIVE MODELING OF FMRI TIME SERIES.

Synthesizing fMRI BOLD signals is challenging due to the complex spatiotemporal dependencies, non-stationarity, and interferences arising from neurophysiological fluctuations. Existing time-series generation is principally based on generative adversarial networks (GANs), variational autoencoders (VAEs), and diffusion-based frameworks.

**GAN-based approaches:** Yoon et al. (2019) proposes TimeGAN by extending GAN framework with an embedding function and a supervised loss to better capture temporal dynamics, successfully preserving both the static and dynamic characteristics of synthetic time-series data. COT-GAN introduces a causality-aware optimal transport cost, further aligning real and synthetic samples over time and reducing time-dependent discrepancy between them (Xu et al., 2020).

**VAE-based approaches:** TimeVAE incorporates temporal components into its encoder-decoder network, improving the interpretability of generated time series. Furthermore, it demonstrates success in reducing overall training time compared to adversarial methods (Desai et al., 2021).

**Diffusion-based approaches:** DiffTime improves time-series generation by applying hard constraints to enforce fixed points and global minima; alongside soft constraints introduce penalties to guide the model towards desired temporal trends (Coletta et al., 2023). DiffWave achieves high-fidelity time-series generation by replacing autoregressive dependencies with a diffusion denoising chain (Kong et al., 2020). More recently, ImagenTime and T2I-Diff demonstrated its capability in modelling long-term time-series benchmarks by converting signals into short-time Fourier transform (STFT) as the image representation, offering an alternative for modelling longer continuous signals using spectral components (Naiman et al., 2024; Tew et al., 2025b). In contrast, the wavelet transform provides multi-resolution bands by using adaptive windows that narrow at high frequencies and widen at low frequencies. These adaptive methods make the wavelet transform better at capturing short transients in continuous signals while still capturing slower trends (Murad et al., 2025; WU et al., 2025).

# C    PROOFS AND DERIVATIONS

## C.1    PROOF OF PROPOSITION 1

*Proof.* The forward-time SPDE (9) in the DCT domain admits the following mode-wise decomposition:

$$dz_t[k] = \eta(t)\,\lambda_k\,z_t[k]\,dt + g(t,k)\,dW_t[k] \tag{18}$$

where $W_t[k]$ is the per-mode standard Wiener process. Subsequently, introduce the variance-preserving (VP) scaling

$$z_t[k] = \alpha(t)\,\tilde{z}_t[k] \tag{19}$$

where $\alpha(t)$ is a scalar applied equally to every mode, and the DCT basis remains unchanged, i.e., the scaled $z_t$ still obeys the heat dissipation SPDE. Substituting this into (18) and applying Itô's lemma gives

$$dz_t[k] = f(t,k)\,z_t[k]\,dt + g(t,k)\,dW_t[k] \tag{20}$$

where we have defined

$$f(t,k) = \frac{\dot{\alpha}(t)}{\alpha(t)} - \eta(t)\,\lambda_k \tag{21}$$

Taking the conditional expectation of the drift term in (20) and integrating with respect to time yields

$$\frac{d}{dt} \mathbb{E}\big[z_t[k] \,|\, z_0[k]\big] = f(t,k) \, \mathbb{E}\big[z_t[k] \,|\, z_0[k]\big]$$

$$\mathbb{E}\big[z_t[k] \,|\, z_0[k]\big] = \int_0^t f(t,k) \, \mu(t,k) \, dt = \alpha(t) \, e^{-\lambda_k \tau(t)} = \mu(t,k) \tag{22}$$

which is exactly the mean schedule defined in (12). From (24), we also have

$$\dot{\mu}(t,k) = f(t,k) \, \mu(t,k) \tag{23}$$

which we will use to derive the standard deviation.

Applying Itô's lemma once again to the square of (20), and taking conditional expectations yields

$$\frac{d}{dt} \mathbb{E}[z_t[k]^2] = 2 \, f(t,k) \, \mathbb{E}\big[z_t[k]^2\big] + g(t,k)^2 \tag{24}$$

Additionally, taking the time-derivative

$$\sigma(t,k)^2 = \mathrm{Var}\big[z_t[k] \,|\, z_0[k]\big] = \mathbb{E}\big[z_t[k]^2\big] - \mu(t,k)^2 \tag{25}$$

and substituting $\dot{\mu} = f(t,k) \, \mu$ from (23), we have

$$\dot{\sigma}^2 = 2 \, f(t,k) \, \sigma^2 + g(t,k)^2 \tag{26}$$

where we use the shorthand notations $\mu$, $\sigma$ and $\dot{\mu}$, $\dot{\sigma}$ for brevity. Since the conditional perturbation kernel is variance–preserving, we also have

$$\sigma(t,k)^2 = 1 - \mu(t,k)^2 \tag{27}$$

Differentiating this gives

$$\dot{\sigma}^2 = -2 \, \mu \, \dot{\mu} = -2 \, f(t,k) \, \mu^2 = -2 \, f(t,k) \, (1 - \sigma^2) \tag{28}$$

Equating (26) and (28) gives

$$g(t,k)^2 = 2 \, \sigma(t,k) \, \big(\dot{\sigma}(t,k) - f(t,k) \, \sigma(t,k)\big) \tag{29}$$

which is exactly (13). This completes the proof. $\qquad\square$

## C.2 PROOF OF PROPOSITION 2

*Proof.* The Gaussian reparameterization trick

$$z_t[k]\big|_{z_0[k]} = \mu(t,k) \, z_0[k] + \sigma(t,k) \, \epsilon \tag{30}$$

follows from the mode-wise conditional perturbation kernel (11) , and its time-derivative gives the conditional vector field (14). Using the results (21), (23) and (29) from the proof of Proposition 1, and substituting (30), we can reformulate the conditional vector field (14) as follows:

$$\begin{aligned}
\frac{dz_t[k]}{dt}\bigg|_{z_0[k]} &= v(z_t \,|\, z_0; t, k) \\
&= \dot{\mu} \, z_0[k] + \dot{\sigma} \, \epsilon \\
&= \frac{\dot{\mu}}{\mu} \, (z_t[k] - \sigma \, \epsilon) + \dot{\sigma} \, \epsilon \\
&= f(t,k) \, \big(z_t[k]\big|_{z_0[k]} - \sigma \, \epsilon\big) + \dot{\sigma} \, \epsilon \\
&= f(t,k) \, z_t[k]\big|_{z_0[k]} + \big(\dot{\sigma} - f(t,k) \, \sigma\big) \, \epsilon \\
&= f(t,k) \, z_t[k]\big|_{z_0[k]} + \frac{1}{2} \, g(t,k)^2 \, \frac{\epsilon}{\sigma} \\
&= f(t,k) \, z_t[k]\big|_{z_0[k]} + \frac{1}{2} \, g(t,k)^2 \, \nabla_{z_t[k]} \log p(z_t[k] \,|\, z_0[k])
\end{aligned} \tag{31}$$

which arrives at the conditional probability flow ODE (15). Here, we again use the shorthand notations for brevity.

Applying the law of the unconscious statistician from (16)

$$\mathbb{E}_{p_{\text{data}}(z_0|z_t)}\big[v(z_t|z_0; t, k) \,|\, z_t\big] \tag{32}$$

to the score $\nabla_{z_t} \log p(z_t \,|\, z_0)$, we have

$$
\begin{aligned}
&\int_{\mathbb{R}} \nabla_{z_t} \log p(z_t \,|\, z_0) \, p_{\text{data}}(z_0|z_t) \, dz_0 \\
&= \int_{\mathbb{R}} \nabla_{z_t} \log p(z_t \,|\, z_0) \, \frac{p(z_t \,|\, z_0) \, p_{\text{data}}(z_0)}{\int_{\mathbb{R}} p(z_t \,|\, z_0) \, p_{\text{data}}(z_0) \, dz_0} \, dz_0 \\
&= \int_{\mathbb{R}} \frac{\nabla_{z_t} p(z_t \,|\, z_0)}{p(z_t \,|\, z_0)} \, \frac{p(z_t \,|\, z_0) \, p_{\text{data}}(z_0)}{p(z_t)} \, dz_0 \\
&= \frac{1}{p(z_t)} \nabla_{z_t} \int_{\mathbb{R}} p(z_t \,|\, z_0) \, p_{\text{data}}(z_0) \, dz_0 \\
&= \frac{1}{p(z_t)} \nabla_{z_t} p(z_t) = \nabla_{z_t} \log p(z_t)
\end{aligned}
\tag{33}
$$

where we have repeatedly apply the log-derivative trick $\frac{1}{p(z)} \nabla p(z) = \nabla \log p(z)$. This gives us the marginal score and the same applies to the drift term $f(t, k) \, z_t[k]|_{z_0[k]}$ in (31), thus completing the proof. □

## D EVALUATION PROTOCOL

### D.1 BASELINES.

**Time-series Generative Models.** Our proposed DSFM model is first assessed in the unconditional setting using standard metrics used by T2I-Diff (Tew et al., 2025b), against seven time-series and time-frequency generative model baselines such as CoT-GAN (Xu et al., 2020), DiffTime (Coletta et al., 2023), DiffWave (Kong et al., 2020), TimeVAE (Desai et al., 2021), TimeGAN (Yoon et al., 2019), Diffusion-TS (Yuan & Qiao, 2024), and T2I-Diff (Tew et al., 2025b). In the conditional setting, we computed the image-domain FID score on subject-specific DCT and DWT image representations (Heusel et al., 2017). To ensure image-to-signal reconstruction quality, we evaluate the time-domain using the context-FID (cFID) score (Jeha et al., 2022).
**fMRI Generative Models.** We further compared DSFM with FC-based GAN models, such as Vanilla-GAN (Goodfellow et al., 2020), 1D-DCGAN (Radford et al., 2015), 2D-DCGAN (Tan et al., 2024b), WGAN and WGAN-GP (Gulrajani et al., 2017).

### D.2 TIME-SERIES METRICS.

We utilize the standard time-series generation metrics from Naiman et al. (2024). We employ the following four metrics and provide their mathematical formulations to ensure comparable evaluation across multiple aspects:

**Discriminative (Disc.) & Predictive score (Pred.).** We adopt the same experimental setup of (Yoon et al., 2019) for both the discriminative and predictive scores. Both the classifier and sequence-prediction model use a two-layer GRU-based architecture. The discriminative score is computed as $|\text{accuracy} - 0.5|$, where lower scores indicate better indistinguishability, and higher scores reflect greater divergence. The predictive score is the mean absolute error (MAE) of the one-step-ahead predictions and the ground-truth values.

**Context-FID score (cFID).** Context-FID score is a time-series adaptation of the image-based Frechet Inception Distance (FID) that measures how close in distribution synthetic data is to the real data in a learned embedding space (Jeha et al., 2022). Instead of image features, it uses a trained encoder called TS2Vec to capture temporal context. Lower scores indicate higher fidelity and have been shown to correlate with better downstream tasks.

**Correlational score (Corr.).** Following (Liao et al., 2020), we first estimate the covariance of the $i$th and $j$th feature of time series as follows:

$$\text{cov}_{i,j} = \frac{1}{T} \sum_{t=1}^{T} X_i^t X_j^t - \left( \frac{1}{T} \sum_{t=1}^{T} X_i^t \right) \left( \frac{1}{T} \sum_{t=1}^{T} X_j^t \right) \tag{34}$$

Then, the correlation score is defined as the average absolute difference between corresponding pairwise correlations in the real and synthetic data:

$$\text{Corr} = \frac{1}{10} \sum_{i,j} \left| \frac{\text{cov}_{i,j}^r}{\sqrt{\text{cov}_{i,i}^r \text{cov}_{j,j}^r}} - \frac{\text{cov}_{i,j}^s}{\sqrt{\text{cov}_{i,i}^s \text{cov}_{j,j}^s}} \right| \tag{35}$$

### D.3 CLASSIFICATION METRICS.

We quantify classification performance using accuracy, precision, recall, F1-score, and the area under the ROC curve, with larger values indicating better performance; their definitions are given in equation 36-equation 40.

$$\text{Accuracy (ACC)} = \frac{\text{TP} + \text{TN}}{\text{TP} + \text{TN} + \text{FP} + \text{FN}} \tag{36}$$

$$\text{Precision (PRE)} = \frac{\text{TP}}{\text{TP} + \text{FP}} \tag{37}$$

$$\text{Recall (REC)} = \frac{\text{TP}}{\text{TP} + \text{FN}} \tag{38}$$

$$\text{F1-score} = 2 \cdot \frac{\text{PRE} \times \text{REC}}{\text{PRE} + \text{REC}} \tag{39}$$

$$\text{ROC} = \int_0^1 \text{TPR}(\tau) \, d\big(\text{FPR}(\tau)\big) \tag{40}$$

## E ADDITIONAL EXPERIMENTAL RESULTS

Table 7: Evaluation of our proposed DSFM with class-conditional (HC vs MDD) generation under varying NFE.

| NFE | Discrete Cosine Transform (DCT) | | Discrete Wavelet Transform (DWT) | | Signal Transform (Time) | |
|---|---|---|---|---|---|---|
| | HC ↓ | MDD ↓ | HC ↓ | MDD ↓ | HC ↓ | MDD ↓ |
| 20 | 5.303±1.890 | 5.352±2.126 | 4.552±0.150 | 4.569±0.203 | 1.703±0.302 | 1.428±0.785 |
| 50 | 5.481±1.892 | 5.580±1.528 | 4.567±0.303 | 4.624±0.100 | 1.327±0.208 | 1.312±0.186 |
| 100 | 5.079±1.507 | 5.386±1.448 | 4.520±0.191 | 4.569±0.134 | 1.237±0.519 | 1.255±0.471 |

### E.1 CONDITIONAL GENERATION QUALITY ACROSS TIME AND FREQUENCY DOMAINS.

Table 7 compares the generative fidelity of our DSFM framework across three domains: frequency (DCT), time-scale (DWT), and the raw time-series representations. Overall, DSFM demonstrates competitive performance in the DWT domain by achieving the lowest FID across HC and MDD subjects with hyperparameter settings of NFE = 100, indicating precise reconstruction of scale-specific BOLD dynamics. Consistently low cFID values in the time domain further confirm that the synthetic signals remain well aligned with in-distribution temporal patterns, outlining that the model is complementary with additional spectral features. In contrast, we also observe that increasing the number of NFE from 20 to 100 consistently reduces error across subjects. These results validate DSFM as an effective time-series-to-image framework for synthesizing biologically plausible, frequency-aligned fMRI signals across representations.

Table 8: Ablation of block size, wavelet basis, normalization strategy and spectral representations (Fourier and wavelet transforms) and type of generative models (flow matching and diffusion).

| Configurations | Context-FID ↓ | Correlational ↓ | Discriminative ↓ | Predictive ↓ |
|---|---|---|---|---|
| 1) $B = 4$, MM | $1.51 \pm 0.41$ | $57.30 \pm 2.89$ | $0.49 \pm 0.00$ | $0.05 \pm 0.00$ |
| 2) $B = 4$, ECS | $0.10 \pm 0.01$ | $18.20 \pm 1.41$ | $0.17 \pm 0.05$ | $0.04 \pm 0.00$ |
| 3) $B = 2$, Haar | $0.12 \pm 0.03$ | $19.70 \pm 3.03$ | $0.13 \pm 0.04$ | $0.04 \pm 0.00$ |
| 4) $B = 4$, Haar | $0.10 \pm 0.01$ | $18.20 \pm 1.41$ | $0.17 \pm 0.05$ | $0.04 \pm 0.00$ |
| 5) $B = 4$, dB-4 | $0.20 \pm 0.10$ | $20.70 \pm 3.25$ | $0.13 \pm 0.04$ | $0.04 \pm 0.00$ |
| 6) Flow Matching, Fourier | $1.00 \pm 0.20$ | $107.90 \pm 14.35$ | $0.46 \pm 0.02$ | $0.05 \pm 0.00$ |
| 7) Flow Matching, Wavelet | $2.19 \pm 0.34$ | $41.88 \pm 3.37$ | $0.29 \pm 0.18$ | $0.04 \pm 0.00$ |
| 8) Diffusion, Wavelet | $3.57 \pm 0.68$ | $44.55 \pm 3.83$ | $0.48 \pm 0.02$ | $0.04 \pm 0.00$ |
| DSFM (Ours) | $0.10 \pm 0.01$ | $18.20 \pm 1.41$ | $0.17 \pm 0.05$ | $0.04 \pm 0.00$ |

### E.2 FULL RESULTS OF ABLATION STUDIES.

Table 8 presents the complete ablation results on the MDD dataset with additional experiments on different spectral representations and type of generative models. In comparison, the Fourier representation achieves a better context-FID score, indicating strength in capturing global distributional characteristics, but it falls short on the other metrics relative to the wavelet representation. This observations exemplify that the time–frequency localization property in wavelet representation is more effective at preserving local and multi-scale structural information. Moreover, the better results of flow matching model with wavelet representation across all metrics suggest a smoother and stable noise-to-data distribution alignment than that of the diffusion-based wavelet approach. Overall, these results demonstrate that DSFM outperformed all the configurations across all time-series generative metrics with the advantage of dual-spectral transformation and spectral flow matching.

### E.3 FULL RESULTS OF CLASSIFICATION PERFORMANCE.

| Method | Train Set | Accuracy ↑ | Recall ↑ | Precision ↑ | F1-Score ↑ | ROC ↑ |
|---|---|---|---|---|---|---|
| W/O Augmentation | Real | $58.90 \pm 2.98$ | $58.90 \pm 2.98$ | $59.56 \pm 2.74$ | $58.39 \pm 3.09$ | $59.00 \pm 2.56$ |
| Vanila-GAN | Real + Synth 1× | $56.90 \pm 1.66$ | $56.90 \pm 1.66$ | $56.40 \pm 2.86$ | $53.68 \pm 3.31$ | $56.29 \pm 1.92$ |
| | Real + Synth 2× | $50.71 \pm 3.69$ | $50.71 \pm 3.69$ | $48.60 \pm 7.23$ | $46.74 \pm 5.18$ | $50.81 \pm 4.13$ |
| | Real + Synth 3× | $58.86 \pm 2.24$ | $58.86 \pm 2.24$ | $59.91 \pm 2.57$ | $57.64 \pm 1.96$ | $58.57 \pm 2.05$ |
| 1D-DCGAN | Real + Synth 1× | $62.94 \pm 2.01$ | $62.94 \pm 2.01$ | $63.43 \pm 2.20$ | $62.23 \pm 2.68$ | $62.71 \pm 2.26$ |
| | Real + Synth 2× | $65.04 \pm 2.02$ | $65.04 \pm 2.02$ | $66.35 \pm 2.13$ | $64.12 \pm 2.10$ | $64.74 \pm 2.08$ |
| | Real + Synth 3× | $58.21 \pm 2.98$ | $58.21 \pm 2.98$ | $55.70 \pm 6.58$ | $52.86 \pm 4.14$ | $57.38 \pm 3.11$ |
| 2D-DCGAN | Real + Synth 1× | $60.78 \pm 4.98$ | $60.78 \pm 4.98$ | $61.30 \pm 5.34$ | $60.01 \pm 5.00$ | $60.33 \pm 5.03$ |
| | Real + Synth 2× | $61.41 \pm 2.59$ | $61.41 \pm 2.59$ | $61.99 \pm 3.73$ | $62.18 \pm 3.29$ | $61.04 \pm 2.79$ |
| | Real + Synth 3× | $62.88 \pm 4.99$ | $62.88 \pm 4.99$ | $63.12 \pm 5.02$ | $62.48 \pm 5.25$ | $62.67 \pm 5.15$ |
| WGAN | Real + Synth 1× | $64.98 \pm 5.54$ | $64.98 \pm 5.54$ | $65.19 \pm 5.34$ | $64.86 \pm 5.61$ | $64.95 \pm 5.39$ |
| | Real + Synth 2× | $60.59 \pm 1.81$ | $60.59 \pm 1.81$ | $60.89 \pm 1.96$ | $60.35 \pm 1.78$ | $60.53 \pm 1.84$ |
| | Real + Synth 3× | $61.83 \pm 3.03$ | $61.83 \pm 3.03$ | $62.27 \pm 3.29$ | $61.44 \pm 2.70$ | $61.58 \pm 2.73$ |
| WGAN-GP | Real + Synth 1× | $66.02 \pm 4.25$ | $66.02 \pm 4.25$ | $66.22 \pm 4.24$ | $65.93 \pm 4.20$ | $65.95 \pm 4.13$ |
| | Real + Synth 2× | $64.76 \pm 4.25$ | $64.76 \pm 4.25$ | $65.67 \pm 4.08$ | $64.23 \pm 4.52$ | $64.73 \pm 4.14$ |
| | Real + Synth 3× | $64.56 \pm 3.18$ | $64.56 \pm 3.18$ | $64.78 \pm 3.17$ | $64.38 \pm 3.15$ | $64.41 \pm 3.08$ |
| T2I-Diff | Real + Synth 1× | $66.87 \pm 3.22$ | $66.87 \pm 3.22$ | $67.06 \pm 3.34$ | $66.83 \pm 3.21$ | $67.26 \pm 6.00$ |
| | Real + Synth 2× | $65.41 \pm 2.37$ | $65.41 \pm 2.37$ | $66.30 \pm 1.67$ | $64.73 \pm 2.80$ | $65.75 \pm 3.22$ |
| | Real + Synth 3× | $66.03 \pm 1.75$ | $66.03 \pm 1.75$ | $66.50 \pm 1.32$ | $65.85 \pm 1.82$ | $66.58 \pm 5.33$ |
| **DSFM (Ours)** | Real + Synth 1× | $70.84 \pm 5.89$ | $70.84 \pm 5.89$ | $70.99 \pm 5.80$ | $70.77 \pm 5.97$ | $71.49 \pm 5.73$ |
| | Real + Synth 2× | $69.58 \pm 3.89$ | $69.58 \pm 3.89$ | $69.75 \pm 3.72$ | $69.43 \pm 3.86$ | $69.91 \pm 4.23$ |
| | Real + Synth 3× | $69.80 \pm 3.13$ | $69.80 \pm 3.13$ | $69.61 \pm 3.02$ | $69.80 \pm 3.13$ | $69.00 \pm 4.29$ |

Table 9: Classification performance (MDD) of different generative models trained on the ground-truth data with an increasing amount of augmented time series data using our proposed model.

| Method | Train Set | Accuracy ↑ | Recall ↑ | Precision ↑ | F1-Score ↑ | ROC ↑ |
|---|---|---|---|---|---|---|
| W/O Augmentation | Real | 64.67 ± 1.56 | 64.67 ± 1.56 | 65.77 ± 2.77 | 64.12 ± 1.39 | 67.28 ± 4.15 |
| Vanila-GAN | Real + Synth 1× | 64.10 ± 3.28 | 64.10 ± 3.28 | 64.33 ± 3.34 | 63.74 ± 3.50 | 65.59 ± 3.83 |
|  | Real + Synth 2× | 64.87 ± 1.05 | 64.87 ± 1.05 | 65.06 ± 1.28 | 64.75 ± 0.93 | 68.07 ± 2.88 |
|  | Real + Synth 3× | 62.94 ± 4.68 | 62.94 ± 4.68 | 62.94 ± 4.69 | 62.91 ± 4.70 | 64.48 ± 5.20 |
| 2D-DCGAN | Real + Synth 1× | 65.63 ± 1.91 | 65.63 ± 1.91 | 66.63 ± 2.44 | 65.26 ± 1.90 | 68.44 ± 1.23 |
|  | Real + Synth 2× | 65.64 ± 1.41 | 65.64 ± 1.41 | 66.37 ± 2.56 | 65.32 ± 1.08 | 67.80 ± 3.32 |
|  | Real + Synth 3× | 64.86 ± 2.87 | 64.86 ± 2.87 | 65.10 ± 2.82 | 64.66 ± 2.96 | 67.92 ± 4.46 |
| TimeGAN | Real + Synth 1× | 66.59 ± 2.50 | 66.59 ± 2.50 | 67.93 ± 1.87 | 65.92 ± 2.92 | 67.49 ± 3.26 |
|  | Real + Synth 2× | 66.24 ± 1.07 | 66.24 ± 1.07 | 66.89 ± 1.37 | 65.84 ± 1.63 | 69.21 ± 2.03 |
|  | Real + Synth 3× | 65.26 ± 1.80 | 65.26 ± 1.80 | 65.37 ± 1.83 | 65.21 ± 1.81 | 64.80 ± 1.81 |
| DiffusionTS | Real + Synth 1× | 66.60 ± 1.41 | 66.60 ± 1.41 | 66.60 ± 1.41 | 66.58 ± 1.41 | 68.85 ± 3.64 |
|  | Real + Synth 2× | 66.02 ± 2.18 | 66.02 ± 2.18 | 66.20 ± 2.01 | 65.98 ± 2.18 | 66.10 ± 2.09 |
|  | Real + Synth 3× | 64.87 ± 1.48 | 64.87 ± 1.48 | 65.10 ± 1.54 | 64.78 ± 1.41 | 66.24 ± 2.92 |
| T2I-Diff | Real + Synth 1× | 69.69 ± 1.55 | 69.69 ± 1.55 | 69.78 ± 1.74 | 69.65 ± 1.53 | 71.88 ± 2.47 |
|  | Real + Synth 2× | 69.49 ± 2.14 | 69.49 ± 2.14 | 69.76 ± 1.90 | 69.31 ± 2.38 | 71.23 ± 2.84 |
|  | Real + Synth 3× | 68.91 ± 1.57 | 68.91 ± 1.57 | 69.22 ± 1.80 | 68.79 ± 1.49 | 68.79 ± 2.40 |
| **DSFM (Ours)** | Real + Synth 1× | 71.54 ± 1.87 | 71.54 ± 1.87 | 73.08 ± 3.00 | 70.98 ± 2.30 | 73.78 ± 3.64 |

Table 10: Classification performance (ABIDE) of different generative models trained on the ground-truth data with an increasing amount of augmented time series data using our proposed model.

Table 9 and 10 presents the complete classification results on the MDD and ABIDE datasets across three augmentation levels. The consistent performance gains at each augmentation level indicate that the synthesized functional conductivity (FC) matrices accurately capture brain connectivity patterns and that the data augmentation strategy significantly improves classifier's generalization to unseen samples. Notably, DSFM achieved better performance even with only a single augmentation level.

## F ADDITIONAL VISUALIZATION

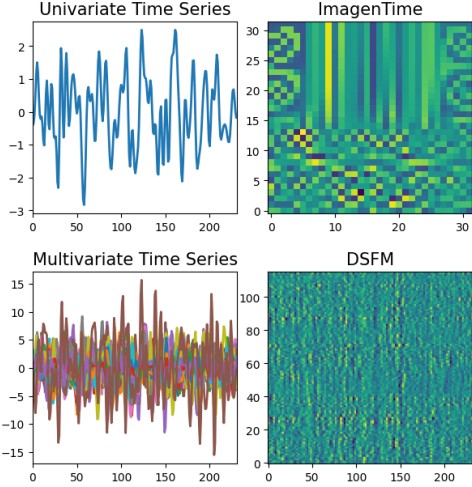

Figure 6: Comparison of univariate and multivariate spectral representations: ImagenTime/T2I-Diff and our proposed DSFM.

### F.1 SPECTRAL IMAGE TRANSFORMATIONS

Figure 6 illustrates the forward and inverse processes of ImagenTime/T2I-Diff and DSFM applied to our proposed fMRI signals. The top row shows an univariate STFT real-valued coefficients, and

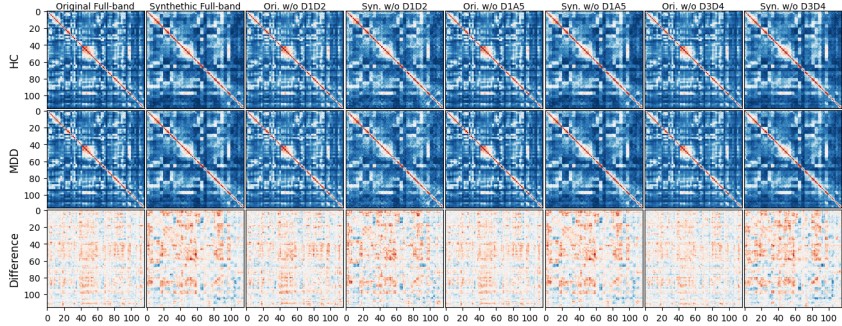

Figure 7: Frequency-specific functional connectivity (FC) matrices for healthy controls (HC) and patients with major depressive disorder (MDD), alongside their differences. The FCs are shown under four different conditions: full-band; removal of the highest-frequency subbands (D1 + D2) and the lowest-frequency component (A5), which both yield the two highest classification scores; and removal of the mid-band subbands (D3 + D4), which produces the greatest deviations and the lowest score.

the bottom row presents a single subband (Detail 1) of multivariate DWT coefficient map. Our framework directly transforms multivariate BOLD signals into a single image representation.

## F.2 FREQUENCY-SPECIFIC FC ANALYSIS

Figure 7 compares the HC and MDD FC matrices against the ground-truth data correlation across different wavelet subbands. Consistent with the full-band correlation, removing the highest-frequency subbands (D1 and D2), or combining D1 with the lowest band (A5) preserves dense edge connections near the main diagonal. In contrast, removing the mid-frequency subbands (D3 and D4) results in sparser connectivity, particularly in the lower-right region of the matrices.

## G COMPUTATIONAL COST

The training of the proposed DSFM required 22 hours, 40 minutes, and 52.698 seconds of wall-clock time, while inference for generating the full samples took 48 minutes and 48.98 seconds with 1x A100 GPU. The model contains 130,844,352 parameters.

## H LIMITATIONS

Currently, DSFM is specially designed for the generation of resting-state fMRI signals. This opens a valuable opportunity to expand our work to other human brain activity signals, such as electroencephalography (EEG), functional near-infrared spectroscopy (fNIRS), and magnetoencephalography (MEG). Our spectral flow matching framework offers flexibility to capture spectral-temporal dynamics of other neural signals with frequency-specific representation.

## I REPRODUCIBILITY STATEMENT

We provide the datasets, source code, and configurations for all key experiments, including instructions on how to preprocess data and train the models at https://github.com/htew0001/DSFM.git.

## J THE USE OF LARGE LANGUAGE MODELS (LLMS)

We used LLMs solely for grammar correction. All ideas, analyses, and results are by the authors.

