# OpenReview forum: "Functional MRI Time Series Generation via Wavelet-Based Image Transform and Spectral Flow Matching for Brain Disorder Identification"
_ICLR.cc/2026/Conference — ICLR 2026 Poster_

### Official Review · Reviewer_UQBR · 2025-10-31

**Soundness:** 4
**Presentation:** 4
**Contribution:** 4
**Rating:** 8
**Confidence:** 5

**Summary:**

This paper presents a method to generate fMRI time-series data. The authors propose a three-step process.
The first step computes a time-frequency representation from the training data using a wavelet transform. Then the transformed signals are then transformed using a 2D Discrete Cosine Transform to further localize the low-frequency BOLD spectral components. Then they use a flow-matching approach that operates in the DCT space that relies on an image generation model (U-ViT) to parameterize and train the model that minimizes the "spectral flow-matching loss" to learn velocity fields in this space. This process further allows to sample the DCT coefficients by integrating the velocity flow, which is then inverse-transformed (both DCT and wavelet) to generate the reconstructed fMRI signals. The authors compare their method with several state of the art models. They also present an application that performs classification between HC and MDD patients.

**Strengths:**

Synthetically generating fMRI time-series data is still an open problem. Most approaches have tackled synthetic data generation in modalities such as ECG, EKG, etc., but fMRI time-series generation is complex and thus still remains challenging. Thus this paper makes a good contribution to this problem.

While the DCT and DWT (more so for time-series) have been applied to fMRI time-series data individually, they have been done so primarily for filtering, preprocessing, noise/motion removal etc type applications. To my knowledge, the joint approach, i.e. first perform DWT for time-frequency analysis and then low-frequency signal compaction using DCT is new, especially for fMRI time-series application.

The idea of recasting flow-matching in the spectral domain instead of relying on diffusion models is somewhat novel. This is especially important for fMRI applications, as 4D time-series data is huge and the deterministic ODE-based flow matching idea is magnitudes in order more efficient than diffusion modeling.

The authors present a method for learning the velocity fields of probability flows directly in the DCT space. They derive the spectral flow-matching loss by diagonalizing the Laplacian operator and then propose the probability flow in the DCT space (proposition 1). This is novel.

The authors have provided the code for their method including experimental results and validation. This is a strength. However, also see the Questions below.

**Weaknesses:**

The main weaknesses in the paper are in the experimental results and aspects related to evaluation of the method on BOLD fMRI time-series applications.

The generated fMRI signals are not validated using any physiological or neuroscientific basis. They are simply validated by context FID scores. Thus the validation is relying on comparing data distributions where the synthetic data is supposed to have come from. This is weak.

Furthermore, using classification accuracy as a metric to discriminate HC against MDD is a complex high-level application, which may mask underlying subtleties of scanner, population, processing methods etc. Thus it is hard to make any judgement on the actual reconstructed time-series signals.

Ablation study is performed on omitting wavelet sub-bands. However, the authors don't comment on the dual-formulation advantage. I.e. is the real gain coming from the spectral representation itself or from flow-matching process, generating the velocity fields in the DCT space.

Only a single dataset (REST-MDD) is used. The dataset contains 250 Healthy Controls (HC) subjects and 227 individuals diagnosed with MDD. This is a much limited dataset to test the method on.

The claim "brain disorder identification" is both pretty general and strong. One could at best say, the method showed improved performance in classification after using the synthetic time-series on the MDD dataset. On that note, the authors don't mention what MDD is. It is supposed to stand for major depressive disorder. Instead they wrongly use the acronym MDD to denote a general brain disorder. Line 88 says: Our results show that DSFM demonstrates strong performance on unconditional and conditional spectral image synthesis, and achieves improvement in brain disorder (MDD) classification. This is an incorrect terminology.

The paper visualizes the average connectivity patterns of real and synthetic connectivity patterns. Without knowing the error bars or any quantitative tests, it is difficult to tell if the synthetic data generation method has worked. Visually, one can see differences in anatomical locations in the connectivity patterns from real and synthetic data and even in their differences. Thus the method is not as accurate, but it is not clear by how much.

For validation purposes, the method does not comment on frequency bands (LF vs HF) that are captured or accurately reflected in the synthetic data. Absence of such low-level measures makes the experimental impact of the contribution, difficult to judge.

The method is compared to several state of the art time-series generation methods. But comparisons to the more recent rectified-flow matching methods are missing. This is important especially as this particular method does rely on a flow-based formulation.

**Questions:**

In the reproducibility statement, the authors have stated, "We provide the datasets, source code, and configurations for all key experiments, including instruc-tions on how to preprocess data and train the models at https://anonymous.4open.science/r/DSFM-
123C" However the link returned "File not Found".

What happens if the DCT step is short-circuited? I.e. the flow velocity fields are learnt on the DWT components?

---

> ### Author Response · Authors · 2025-12-04
> **Weakness 1,2,3**
>
> # Weakness 1,2:
> **The generated fMRI signals are not validated using any physiological or neuroscientific basis. They are simply validated by context FID scores. Thus the validation is relying on comparing data distributions where the synthetic data is supposed to have come from. This is weak.**
>
> **Furthermore, using classification accuracy as a metric to discriminate HC against MDD is a complex high-level application, which may mask underlying subtleties of scanner, population, processing methods etc. Thus it is hard to make any judgement on the actual reconstructed time-series signals.**
>
> # Response:
> We thank the reviewer for pointing out that our initial submission lacked physiological or neuroscientific validation of the generated BOLD signals. In the revised manuscript, we have added a neurophysiological plausibility analysis to directly address this concern. First, we compute and compare the resting-state hemodynamic response function (rsHRF) between real and synthetic signals (see Fig. 4, page 9). The close overlap of the estimated rsHRF curves demonstrates that DSFM preserves canonical temporal dynamics of the hemodynamic process rather than merely matching distributional statistics. Second, we include a power spectral density (PSD) analysis (Fig. 4, page 9), which shows that the generated signals accurately reproduce the characteristic spectral structure of fMRI BOLD activity, including dominant low-frequency oscillations and the decay profile across frequencies. Together, these analyses provide direct physiological evidence that DSFM reconstructs biologically meaningful temporal and spectral patterns in fMRI time series. All corresponding results and explanations are now included in the revised manuscript.
>
>
> # Weakness 3:
> **Ablation study is performed on omitting wavelet sub-bands. However, the authors don't comment on the dual-formulation advantage. I.e. is the real gain coming from the spectral representation itself or from flow-matching process, generating the velocity fields in the DCT space.**
>
> # Response:
> We thank the reviewer for raising the important question of whether transforming the fMRI signal into the frequency domain is strictly necessary, and whether a latent-space flow matching baseline (e.g., transformer encoders applied directly to time-domain ROI sequences without DWT/DCT) could yield comparable benefits. In our revised manuscript, Table 2 compares diffusion-based time-domain models (e.g., Diffusion-TS) with our proposed spectral-domain approach on the MDD dataset. Both the generation quality and downstream classification performance strongly indicate that operating in the spectral domain is necessary for high-fidelity modeling.Unlike time-domain approaches, the dual DWT-DCT representation explicitly addresses the non-stationarity of BOLD signals by isolating transient fluctuations (wavelets) and compressing slow-varying oscillatory structure (DCT). Diffusion-TS, which does not work in the spectral domain, struggles to capture these non-stationary temporal dynamics, leading to suboptimal conditional generation quality and weaker classification performance. These results demonstrate that spectral-domain modeling offers substantive advantages over time-domain latent embeddings, and we have clarified this rationale in the revised text.

---

> ### Author Response · Authors · 2025-12-04
> **Weakness 4**
>
> # Weakness 4:
> **Only a single dataset (REST-MDD) is used. The dataset contains 250 Healthy Controls (HC) subjects and 227 individuals diagnosed with MDD. This is a much limited dataset to test the method on. The claim "brain disorder identification" is both pretty general and strong. One could at best say, the method showed improved performance in classification after using the synthetic time-series on the MDD dataset. On that note, the authors don't mention what MDD is. It is supposed to stand for major depressive disorder. Instead they wrongly use the acronym MDD to denote a general brain disorder. Line 88 says: Our results show that DSFM demonstrates strong performance on unconditional and conditional spectral image synthesis, and achieves improvement in brain disorder (MDD) classification. This is an incorrect terminology.**
>
> # Response:
> We thank the reviewer for highlighting the limited dataset scope and the incorrect usage of the acronym MDD in the original manuscript. In the revised version, we now define MDD (Major Depressive Disorder) upon first mention and ensure that we no longer use “MDD” as a stand-in for general “brain disorder,” but instead clearly state that our classification experiments are specific to Major Depressive Disorder. To address concerns regarding dataset diversity, we have expanded our empirical evaluation beyond the REST-meta-MDD dataset. In addition to the NetSim dataset used for unconditional time-series generation, we now include experiments on the ABIDE dataset, which differs substantially in subject population, pathology (autism rather than depression), and anatomical parcellation. We use stratified 5-fold cross-validation for the MDD dataset with AAL116 and 10-fold cross-validation for ABIDE using the Schaefer atlas, enabling us to assess robustness across heterogeneous datasets and parcellation schemes. These updates, along with expanded analyses presented in Tables 2, 3, and 6 and Fig. 4, have been incorporated into Section 3.1 of the revised manuscript. We believe these additions demonstrate that our method performs consistently across one simulated dataset and two large real-world clinical datasets, while broader multi-disorder evaluations will be an important direction for future work.

---

> ### Author Response · Authors · 2025-12-04
> **Weakness 5,6**
>
> # Weakness 5:
> **The paper visualizes the average connectivity patterns of real and synthetic connectivity patterns. Without knowing the error bars or any quantitative tests, it is difficult to tell if the synthetic data generation method has worked. Visually, one can see differences in anatomical locations in the connectivity patterns from real and synthetic data and even in their differences. Thus the method is not as accurate, but it is not clear by how much.**
>
> # Response:
> We thank the reviewer for pointing out the need for quantitative uncertainty estimates in our connectivity visualizations. In the revised manuscript, we have updated Figure 5 to include error bars for both real and synthetic connectivity patterns. The resulting confidence ranges demonstrate that the synthetic FCs produced by DSFM closely resemble the real data, with deviations well within the expected variability across subjects. In addition to the visual analysis, we have included a quantitative evaluation of FC similarity in Table 6, where DSFM achieves higher Pearson correlations with real FC matrices than GAN-based baselines. This indicates that our method more accurately preserves both individual connectivity edges and the broader network topology. Together, these results show that DSFM not only produces plausible pairwise connectivity patterns but also captures coherent higher-order interdependencies across ROIs, providing a more faithful reconstruction of functional brain networks than existing generative models.
>
> # Weakness 6:
> **For validation purposes, the method does not comment on frequency bands (LF vs HF) that are captured or accurately reflected in the synthetic data. Absence of such low-level measures makes the experimental impact of the contribution, difficult to judge. The method is compared to several state of the art time-series generation methods. But comparisons to the more recent rectified-flow matching methods are missing. This is important especially as this particular method does rely on a flow-based formulation.**
>
> # Response:
> We thank the reviewer for raising the important question of whether specific frequency bands (e.g., low-frequency vs. high-frequency components) are faithfully captured in the synthetic data. In response, we have added a power spectral density (PSD) analysis in Figure 4, which demonstrates that the generated BOLD signals preserve the characteristic spectral structure of real fMRI data, including the dominance of low-frequency oscillations and the expected decay across higher frequencies. This provides direct evidence that DSFM accurately reconstructs the frequency content of the underlying physiological signal. Regarding comparisons to rectified-flow matching methods, to the best of our knowledge, DSFM is the first flow-matching framework developed for fMRI time-series generation. As such, no prior rectified-flow model exists in this domain. Nevertheless, we include Diffusion-TS, which shares the same transformer backbone but is trained under a diffusion formulation rather than flow matching, providing a meaningful and rigorous baseline for comparing our generative process. These additions and clarifications are now included in the revised manuscript.

---

> ### Author Response · Authors · 2025-12-04
> **Question 1**
>
> # Question 1:
> **In the reproducibility statement, the authors have stated, "We provide the datasets, source code, and configurations for all key experiments, including instruc-tions on how to preprocess data and train the models at https://anonymous.4open.science/r/DSFM- 123C" However the link returned "File not Found".**
>
> # Response:
> We thank the reviewer for pointing out that the reproducibility link in our initial submission was not functioning. We have updated the link in the revised manuscript to ensure full accessibility. In addition, the source code used in all experiments has been attached in the supplementary materials. After acceptance, we will also publicly release the full codebase together with all preprocessing scripts and preprocessed data to further support reproducibility.

---

### Official Review · Reviewer_7U7B · 2025-11-01

**Soundness:** 3
**Presentation:** 2
**Contribution:** 3
**Rating:** 4
**Confidence:** 4

**Summary:**

The paper proposes Dual-Spectral Flow Matching (DSFM) for fMRI time-series generation. It converts BOLD signals into a dual-spectral representation using DWT and DCT, then applies a standard flow-matching model to synthesize time-series data for data augmentation on a depression vs control classification task.

**Strengths:**

1. Combining DWT and DCT with flow-matching for fMRI synthesis is a creative approach.

2. The demonstrated use of fMRI data augmentation for enhancing downstream classification performance is interesting, and the proposed application appears promising.

**Weaknesses:**

Major:

1. The experiment scope is a bit limited. All experiments are done on a single dataset (REST-meta-MDD). There is no cross-dataset or cross-site validation, so the generalization and robustness claims are weak.

2. The paper repeatedly mentions that the proposed spectral-domain flow matching is more efficien, but this claim is not supported by theoretical analysis or empirical evidence. It would strengthen the work if the authors could elaborate on what “efficiency” specifically refers to (e.g., faster sampling, fewer function evaluations, or reduced computational cost) and provide quantitative comparisons to substantiate this point.

3. The paper does not include experiments that clarify whether transforming fMRI signals into the frequency domain is strictly necessary. A direct comparison with a latent-space flow matching baseline, i.e., encoding fMRI ROI time series into latent embeddings with transformer encoder without explicit DWT/DCT transforms, would help determine whether the spectral transforms genuinely contribute to the observed improvements in performance or efficiency, or if similar gains could be achieved through compact latent representations.

4. The authors note that their primary goal is not to achieve the best sample quality metrics but rather to improve cFID and conditional modeling of spatiotemporal patterns. However, this statement seems to suggest that the proposed DSFM does not outperform recent baselines (e.g., Diffusion-TS) in unconditional generation quality (Table 1). Can the author elaborate more on this, why cFID is more important than other metrics in this scenario?


Minor:
1. For the connectivity matrices in Figure 4, would be more informative to include colorbar to show the value scale.

2. There might be format error at line 291 and line 308

3. While the technical content is interesting, the overall presentation could be improved for clarity, especially the method and theory parts.

4. Potential typo at line 136-137 "Thus forming a full wavelet..."

5. There is another recent baseline for fMRI synthesis [1]. Although it is not open-sourced yet, it would still be worthwhile to discuss it in the related work section to provide a more complete context

[1] Synthesizing Realistic fMRI: A Physiological Dynamics-Driven Hierarchical Diffusion Model for Efficient fMRI Acquisition, ICLR 2025

**Questions:**

1. The paper fix the wavelet type to Haar. Did author experiment with different wavelet bases (e.g., Daubechies, Coiflet, Symlet)? Since different wavelets have distinct time–frequency localization and smoothness properties, this omission makes the transform choice look arbitrary.
2. What's the rationale of using zig zag flattening rather than normal flattening in Figure 1(step 3)? What would be the difference in model performance
3. What could be the possible reason that, in Figure 4, the synthetic connectivity matrices appear more clustered and uniformly distributed compared to the real ones?

---

> ### Author Response · Authors · 2025-12-04
> **Major Weakness 1**
>
> # Weakness:
> **The experiment scope is a bit limited. All experiments are done on a single dataset (REST-meta-MDD). There is no cross-dataset or cross-site validation, so the generalization and robustness claims are weak.**
>
> # Response:
> We thank the reviewer for highlighting this important point. Our experiments primarily use the Netsim dataset, a widely adopted benchmark from the FMRIB Analysis Group for evaluating unconditional time-series generation. To further demonstrate generalization, we also include experiments on a real-world Major Depressive Disorder (MDD) dataset, which evaluates class-conditional generation. We fully agree that evaluating publicly available datasets is valuable. Therefore, we additionally incorporate the ABIDE dataset, using a different parcellation scheme to assess the robustness of our model across heterogeneous datasets and anatomical parcellations. Specifically, we employ stratified 5-fold cross-validation for the MDD dataset with AAL 116 Atlases and 10-fold cross-validation for the ABIDE dataset with schaffer parcellation, providing extensive robustness assessments. The details of the dataset and the results are added in section 3.1 in our revised manuscript.

---

> ### Author Response · Authors · 2025-12-04
> **Major Weakness 2**
>
> # Weakness:
> **The paper repeatedly mentions that the proposed spectral-domain flow matching is more efficien, but this claim is not supported by theoretical analysis or empirical evidence. It would strengthen the work if the authors could elaborate on what “efficiency” specifically refers to (e.g., faster sampling, fewer function evaluations, or reduced computational cost) and provide quantitative comparisons to substantiate this point.**
>
> # Response:
> We thank the reviewer for requesting clarification on the term "efficiency” in the context of spectral-domain flow matching. In our revision, we define efficiency in terms of sampling complexity, measured by the number of function evaluations (NFEs) required to generate a sample. Diffusion-based baselines such as Diffusion-TS rely on multi-step reverse-time SDE solvers and typically require thousands of NFEs (e.g., 10-12k steps) due to their iterative noise-removal procedure.
>
> In contrast, our DSFM framework operates in a compact dual spectral representation where energy is concentrated in low-order DCT modes, enabling the learned flow field to produce smooth transport trajectories that can be integrated using only 20–100 ODE steps (see Table 7), while still achieving SOTA conditional generation and classification performance (see Table 3). Moreover, unlike image-diffusion pipelines such as T2I-Diff. DSFM performs generation directly in an energy-ordered spectral space with substantially reduced dimensionality, lowering both the number of integration steps and computational cost.

---

> ### Author Response · Authors · 2025-12-04
> **Major Weakness 3**
>
> # Weakness:
> **The paper does not include experiments that clarify whether transforming fMRI signals into the frequency domain is strictly necessary. A direct comparison with a latent-space flow matching baseline, i.e., encoding fMRI ROI time series into latent embeddings with transformer encoder without explicit DWT/DCT transforms, would help determine whether the spectral transforms genuinely contribute to the observed improvements in performance or efficiency, or if similar gains could be achieved through compact latent representations.**
>
> # Response:
> We thank the reviewer for raising the important question of whether transforming the fMRI signal into the frequency domain is strictly necessary, and whether a latent-space flow matching baseline (e.g., transformer encoders applied directly to time-domain ROI sequences without DWT/DCT) could yield comparable benefits.
>
> In our manuscript, Table 2 compares diffusion-based time-domain models, Diffusion-TS with our proposed spectral-domain approach on the MDD dataset. Both the generation quality and downstream classification performance strongly indicate that operating in the spectral domain is necessary for high-fidelity modeling. Unlike time-domain approaches, the dual spectral representation explicitly addresses the non-stationarity of BOLD signals by isolating transient fluctuations (wavelets) and compressing slow-varying oscillatory structure (DCT). Diffusion-TS, which does not work in the spectral domain, struggles to capture these non-stationary temporal dynamics, leading to suboptimal conditional generation quality and weaker classification performance.
>
> These results demonstrate that spectral-domain modeling offers substantive advantages over time-domain latent embeddings.

---

> ### Author Response · Authors · 2025-12-04
> **Major Weakness 4**
>
> # Weakness:
> **The authors note that their primary goal is not to achieve the best sample quality metrics but rather to improve cFID and conditional modeling of spatiotemporal patterns. However, this statement seems to suggest that the proposed DSFM does not outperform recent baselines (e.g., Diffusion-TS) in unconditional generation quality (Table 1). Can the author elaborate more on this, why cFID is more important than other metrics in this scenario?**
>
> # Response:
> We thank the reviewer for asking for clarification on the importance of cFID. In our work, we prioritize cFID because it is specifically designed to evaluate temporal generative fidelity in time-series data.
>
> Context-FID adapts the image-based Frechet Inception Distance to the time-series domain by replacing Inception features with representations learned by TS2Vec, a self-supervised encoder that captures temporal context and local–global dependencies. This makes cFID particularly relevant for fMRI, where the signal of interest lies in its temporal dynamics, non-stationarity, and context-dependent fluctuations across ROIs.
>
> Lower cFID scores indicate that the generated sequences are close to the real BOLD signals in this learned embedding space, and prior work shows that cFID correlates strongly with downstream task performance (see DiffusionTS). Although DSFM does not achieve the best unconditional NetSim generation scores in Table 1, it consistently achieves superior conditional generation fidelity and downstream classification accuracy  in MDD and ABIDE datasets (Tables 2–3), which are aligned with our main objective of modeling class-dependent spatiotemporal patterns rather than optimizing unconditional sample quality.

---

> ### Author Response · Authors · 2025-12-04
> **Question 1**
>
> # Question:
> **The paper fix the wavelet type to Haar. Did author experiment with different wavelet bases (e.g., Daubechies, Coiflet, Symlet)? Since different wavelets have distinct time–frequency localization and smoothness properties, this omission makes the transform choice look arbitrary.**
>
> # Response:
> We thank the reviewer for highlighting the importance of evaluating different wavelet bases. The choice of a mother wavelet is closely related to the characteristics of the BOLD signal. For example, Haar wavelets are well suited for signals with piecewise-constant structures and sharp transitions, whereas smoother wavelets such as Daubechies, Coiflet, or Symlet families are typically preferred for signals exhibiting higher-order smoothness. In practice, unless the signal exhibits sharply distinctive features, the performance difference across wavelet families is often consistent.
>
> To address the reviewer’s suggestion, we have now included additional experiments using the db-4 wavelet basis (see Table 6). The results show that Haar wavelets yield slightly better generative performance than the smoother db-4 wavelet. This shows that the piecewise-constant basis functions of Haar capture the coarse, low-frequency dominant structure of fMRI BOLD signals more effectively, whereas smoother wavelets may oversmooth subtle temporal variations in our proposed fMRI dataset. These findings support our choice of Haar as a practical and empirically effective in our proposed MDD dataset.

---

> ### Author Response · Authors · 2025-12-04
> **Question 2**
>
> # Question:
> **What's the rationale of using zig zag flattening rather than normal flattening in Figure 1(step 3)? What would be the difference in model performance.**
>
> # Response:
>
> We thank the reviewer for highlighting the rationale behind using zigzag flattening instead of a normal flattening in Figure 1 (Step 3). This design choice is essential for preserving the frequency structure of the DCT-transformed fMRI representation.
>
> A standard flattening operation preserves only spatial adjacency, producing a 1D sequence in which low-, mid-, and high-frequency DCT coefficients are interleaved according to arbitrary spatial positions. Because DCT coefficients encode frequency content rather than spatial layout, such a flattening disrupts the spectral organization of the block and offers no meaningful grouping of frequency bands. This leads to a coefficient vector whose ordering is not aligned with the intrinsic structure of the signal, making it more difficult for the generative model to learn coherent spectral dependencies.
>
> In contrast, the zigzag scanning algorithm is a principled and widely used method (e.g. JPEG) for linearizing DCT blocks. It traverses coefficients diagonally from low to high frequency, starting at the DC component and moving progressively through increasing frequency bands. This produces a frequency-ordered vector in which low-frequency components (dominant signals) appear first, where high-frequency components (noise) are placed later in the sequence. This ordering preserves the spectral hierarchy of the block and simplifies the model’s learning of multi-scale temporal structure, which is beneficial in fMRI signal generation.
>
> For these reasons, and because fMRI BOLD signals exhibit strong low-frequency dominance spectral structure, we adopt zigzag scanning exclusively for the fMRI scenario to ensure that the DCT representation remains physiologically meaningful and statistically coherent for the generative model.

---

> ### Author Response · Authors · 2025-12-04
> **Question 3**
>
> # Questions:
> **What could be the possible reason that, in Figure 4, the synthetic connectivity matrices appear more clustered and uniformly distributed compared to the real ones?**
>
> # Response:
> We thank the reviewer for pointing out that the synthetic connectivity matrices in Figure 4 appear more clustered and uniformly distributed than the real ones. We conducted ablation analyses (see Table 6), we found that this effect was primarily caused by the MinMax normalization used in our initial preprocessing pipeline. MinMax rescales each DCT/wavelet coefficient independently, which broadens the dynamic range of high-frequency components that naturally carry low energy in fMRI BOLD signals. This distortion leads the model to overemphasize these high-frequency coefficients, producing synthetic connectivity matrices that appear more uniform and clustered than the real data.
>
> To address this, we introduced Entropy-Consistent Scaling (ECS), which normalizes the DCT coefficients using a percentile-trimmed bound derived from the dominant lowest-frequency component. This preserves the global spectral energy structure and avoids artificially amplifying high-frequency noise. As shown in our ablation results in Table 6, ECS yields substantially better generation quality, reflected in improved cFID and correlation scores.
>
> We have updated the connectivity matrices (see Figure 4) in the revised manuscript using ECS-normalized data. The new results show that the synthetic connectivity matrices exhibit more realistic variability and structure, closely matching the distributional characteristics of the real connectivity matrices.

---

### Official Review · Reviewer_YXrp · 2025-11-01

**Soundness:** 2
**Presentation:** 3
**Contribution:** 2
**Rating:** 4
**Confidence:** 4

**Summary:**

This paper proposes a framework named Dual-Spectral Flow Matching (DSFM) for fMRI BOLD signal generation. The method jointly constructs dual-spectral representations through Discrete Wavelet Transform (DWT) and Discrete Cosine Transform (DCT), then introduces Spectral Flow Matching (SFM) in the DCT domain to achieve fMRI generation by modeling the heat diffusion process in the frequency domain.

**Strengths:**

Innovative dual-spectral architecture:
The paper introduces a dual-spectral generative framework combining DWT (Discrete Wavelet Transform) and DCT (Discrete Cosine Transform).
This design elegantly captures both temporal–frequency representations and spectral sparsity of fMRI signals, offering a novel perspective distinct from previous time-domain or FC-based generation approaches.

Solid theoretical derivation:
The work provides a rigorous and mathematically sound derivation linking stochastic partial differential equations (SPDE) to frequency-domain probability flow ODEs, establishing a coherent theoretical foundation for the proposed spectral flow matching.
The formulation is complete and clearly connects to existing flow-matching paradigms.

The experimental design is comprehensive, and the analysis is detailed.

**Weaknesses:**

Lack of cross-dataset validation and statistical significance tests:
Experiments are conducted only on the REST-meta-MDD dataset, without evaluation on other publicly available fMRI datasets such as ABIDE [1] and HCP [2].
This raises concerns about the model’s generalizability across acquisition protocols and subject populations.
Moreover, the reported improvements lack statistical significance testing, making it difficult to assess the robustness of the observed gains.


[1] Heinsfeld A S, Franco A R, Craddock R C, et al. Identification of autism spectrum disorder using deep learning and the ABIDE dataset[J]. NeuroImage: clinical, 2018, 17: 16-23.

[2] Smith S M, Beckmann C F, Andersson J, et al. Resting-state fMRI in the human connectome project[J]. Neuroimage, 2013, 80: 144-168.

Limited physical and physiological interpretability:
Although the proposed DSFM introduces a dual-spectral flow matching framework with physically inspired dynamics, its neuroscientific meaning remains limited. The model is clean and physically inspired from a signal engineering perspective; however, it lacks true mechanistic grounding in neuroscience. Although the paper claims to generate “physiologically plausible” fMRI signals, no quantitative evidence is provided to support this claim. Metrics such as power spectral density (PSD) alignment, hemodynamic response function (HRF) consistency, or spatiotemporal smoothness are not evaluated. Including these analyses would substantially strengthen the physiological credibility of the proposed approach. In essence, DSFM produces spectrally and statistically plausible fMRI-like time series rather than neurodynamically meaningful brain processes. I would encourage the authors to further elaborate on the neuroscientific significance of generating such fMRI signals, for example, what new understanding or insight about brain dynamics can be obtained from this generation process, or how such generated fMRI signals could be practically useful?

Overly dense pipeline illustration:
Figure 1 is visually overloaded. Several components such as "Reorder & Truncation", "SNR scale + Add Noise" and "Zig Zag Flattening" are insufficiently explained in the text, which reduces clarity. Simplifying or modularizing the figure, or adding explicit textual references, would improve readability.

**Questions:**

Check the weakness

---

> ### Author Response · Authors · 2025-12-04
> **Weakness 1**
>
> # Weakness
> **Lack of cross-dataset validation and statistical significance tests: Experiments are conducted only on the REST-meta-MDD dataset, without evaluation on other publicly available fMRI datasets such as ABIDE [1] and HCP [2]. This raises concerns about the model’s generalizability across acquisition protocols and subject populations. Moreover, the reported improvements lack statistical significance testing, making it difficult to assess the robustness of the observed gains.**
>
> # Response
>
> We thank the reviewer for highlighting this important point. Our experiments primarily use the Netsim dataset, a widely adopted benchmark from the FMRIB Analysis Group for evaluating unconditional time-series generation. To further demonstrate generalization, we also include experiments on a real-world Major Depressive Disorder (MDD) dataset, which evaluates class-conditional generation. We fully agree that evaluating publicly available datasets is valuable. Therefore, we additionally incorporate the ABIDE dataset, using a different parcellation scheme to assess the robustness of our model across heterogeneous datasets and anatomical parcellations. Specifically, we employ stratified 5-fold cross-validation for the MDD dataset with AAL 116 Atlases and 10-fold cross-validation for the ABIDE dataset with schaffer parcellation, providing extensive robustness assessments. The details of the dataset and the results are added in section 3.1 in our revised manuscript.

---

> ### Author Response · Authors · 2025-12-04
> **Weakness 2**
>
> # Weakness
> **Limited physical and physiological interpretability: Although the proposed DSFM introduces a dual-spectral flow matching framework with physically inspired dynamics, its neuroscientific meaning remains limited. The model is clean and physically inspired from a signal engineering perspective; however, it lacks true mechanistic grounding in neuroscience. Although the paper claims to generate “physiologically plausible” fMRI signals, no quantitative evidence is provided to support this claim. Metrics such as power spectral density (PSD) alignment, hemodynamic response function (HRF) consistency, or spatiotemporal smoothness are not evaluated. Including these analyses would substantially strengthen the physiological credibility of the proposed approach. In essence, DSFM produces spectrally and statistically plausible fMRI-like time series rather than neurodynamically meaningful brain processes. I would encourage the authors to further elaborate on the neuroscientific significance of generating such fMRI signals, for example, what new understanding or insight about brain dynamics can be obtained from this generation process, or how such generated fMRI signals could be practically useful?**
>
> # Response:
> We thank the reviewer for highlighting the importance of physical and physiological interpretability in evaluating generative fMRI models. We fully agree that beyond statistical plausibility, neuroscientific relevance is essential for assessing the fidelity of synthetic BOLD signals.
>
> To address this, we have added neurophysiological analyses in the revised submission. Specifically, we now include resting-state hemodynamic response function (rsHRF) consistency and power spectral density (PSD) alignment between real and synthetic BOLD signals (see Figure 4, Page 9). These analyses quantify whether the generated data preserve (i) the canonical temporal dynamics of the hemodynamic process and (ii) the low-frequency characteristic of resting-state fMRI. Both HRF and PSD curves exhibit close overlap, demonstrating that DSFM captures physiologically meaningful signal properties rather than only matching distributional statistics.
>
> Furthermore, we provide functional connectivity (FC) evaluations (see Table 7, Page 9), which assess the preservation of network-level organization, including edge-wise connectivity, node strength and edges betweenness centrality. These results complement the HRF/PSD analyses by showing that DSFM not only reconstructs local temporal hemodynamics, but also recovers higher-order network structure for neuroscientific downstream applications.
>
> Most importantly, our method is practically useful (see Tables 3 and 4) that  incorporating DSFM-generated samples consistently improves downstream classification performance, demonstrating its value for data augmentation, especially in resource-intensive nature that is common in neuroimaging. Beyond disorder identification, synthetic fMRI can benefit several applications, including (i) enhancing training stability for disease prediction tasks. (ii) supporting cross-site harmonization through augmentation of under-represented acquisition domains. (iii) enabling robust evaluation when real data are scarce or imbalanced.

---

> ### Author Response · Authors · 2025-12-04
> **Weakness 3**
>
> # Weakness
> **Overly dense pipeline illustration: Figure 1 is visually overloaded. Several components such as "Reorder & Truncation", "SNR scale + Add Noise" and "Zig Zag Flattening" are insufficiently explained in the text, which reduces clarity. Simplifying or modularizing the figure, or adding explicit textual references, would improve readability.**
>
> # Response:
> We thank the reviewer for pointing out that the figures are overloaded. We acknowledge  that Figure 1 is visually dense and that several steps were not clearly referenced in the text.  In the revision, we reorganize Figure 1 into clearer modules (preprocessing, dual-spectral transform, and flow-matching forward, sampling, and downstream task). We believe that these changes will greatly improve clarity.

---

### Official Review · Reviewer_btKb · 2025-11-01

**Soundness:** 2
**Presentation:** 3
**Contribution:** 3
**Rating:** 2
**Confidence:** 3

**Summary:**

This paper proposes a novel approach to fMRI time-series generation based on dual spectral transformations and a learned flow in spectral space. The authors detail the spectral transforms and the flow’s training procedure, and they experimentally validate the method on a single dataset.

**Strengths:**

1/ The subject of study is of utmost interest.

2/ The approach is well described, and most technical details are provided. Figure 1 is an excellent summary.

3/ The range of evaluation metrics and the ablation study make the validation comprehensive (apart from the limited number of datasets).

**Weaknesses:**

Weaknesses summary

Some rationale for the design is missing, and the approach lacks comprehensive empirical validation across additional datasets.

Major concerns

1/ The choice to work in the spectral domain and to cascade spectral transforms is not clearly justified. I am trying to understand why you chose the DCT after a wavelet transform. Apart from the diagonalization of the Laplacian and the simpler truncation it enables, why introduce a second spectral projection and why this one?

2/ If B is too large, it will squash multiple ROIs into a single patch, which in turn will make it impossible for the flow to learn the interactions between those ROIs (since they are collapsed). This is not specifically discussed in the paper, and the experimental values for B are not provided.

3/ In the contributions, you state that you "forming a unified dual-spectral image transform to capture both global and local spatiotemporal and spectral features for fMRI BOLD signal generation"; It is not clear what specifically differentiates your method from prior approaches in capturing ROI-to-ROI interactions (see 2/). Why should these interactions be better captured in spectral space rather that in the time domain? Note that classical fMRI analysis choose to capture the ROIs' interplays in the time domain.

4/ Truncation is crucial since it removes part of the information (note that, at this stage, most artifacts should already have been filtered out by preprocessing); yet it is not specifically discussed in the paper, and the experimental values are not provided. Additionally, for short TRs (< 0.7 s), it remains unclear whether truncation would not be detrimental.

5/ Lines 60 to 63: It is not clear why your approach, which operates in the spectral domain, differs from methods that operate on preprocessed time series in its ability to filter out confounds; the preprocessing pipeline should have already remove cardiac pulsations, respiratory cycles, and motion-induced artifacts. Please clarify which components remain to be removed (see 4/).

6/ The main issue with this paper is the lack of an exhaustive experimental validation. It should be evaluated on at least half a dozen datasets, spanning different TRs, spatial resolutions, and tasks (e.g., multiclass subject prediction).

Minor comments

1/ In the abstract, "preserves key physiological brain dynamics" is an over-claims evidence.

2/ There are numerous typos, for example: "we stitches the patches", "Fig. 1 provide an overview", etc

3/ Some acronyms are undefined or defined too late, for example: MDD, STFT, etc. Even for some common acronyms, a brief reminder would improve readability

4/ Please cite Kawahara et al., 2017 when mentioning BrainNetCNN so the provenance is clear.

5/ The results are somewhat ambiguous: the proposed approach is not SOTA in terms of "generation quality" (Table 1), yet it achieves SOTA prediction performance (Table 3). This discrepancy is surprising and is not discussed.

Grading explanation

The paper lacks comprehensive empirical validation, and some key rationales for the proposed approach are missing.

**Questions:**

1/ Why choose a heat dissipation process? Does avoiding conventional isotropic diffusion help capture interactions between patches (~ ROIs) more effectively?

2/ At a general level, the paper seems to suggest that learning a complex generative distribution (and sampling from it) is easier than discriminating between two of its marginals. This is not a typical results (cf. On Discriminative vs. Generative classifiers: A comparison of logistic regression and naive Bayes), can you elaborate?

3/ Please report the computational cost of your approach: approximate training time, hardware used, and runtime for sample generation.

4/ How do you denormalize the wavelet coefficients (c.f. "We further perform componentwise normalization to accentuate"). Please detail the inverse step.

5/ Do you have an explanation for the discrepancy between Table 1 and Table 3 (non-SOTA generation quality vs. SOTA prediction performance)?

---

> ### Author Response · Authors · 2025-12-04
> **Major Weakness  1**
>
> # Weakness:
> **1/ The choice to work in the spectral domain and to cascade spectral transforms is not clearly justified. I am trying to understand why you chose the DCT after a wavelet transform. Apart from the diagonalization of the Laplacian and the simpler truncation it enables, why introduce a second spectral projection and why this one?**
>
> # Response:
> We thank the reviewer for raising the important question regarding our motivation for working in the spectral domain and for cascading spectral transforms. Our design is inspired by cepstral analysis, where a second-order transform is applied to a spectral representation in order to extract slowly varying structure in the spectrum itself.
>
> In our framework, the wavelet transform producing a multiscale, time–frequency decomposition that isolates local transient dynamics and preserves the nonstationary nature of fMRI BOLD signals. The subsequent DCT is then applied to these wavelet subband coefficients to capture smooth cross-frequency structure and to compact the energy into a small number of low-order cosine components. This yields a representation in which physiologically meaningful low-frequency trends are emphasized while high-frequency noise is naturally attenuated.
>
> Conceptually, this DWT to DCT cascade reveals structure within the spectrum that is not accessible through a single time–frequency transform alone. This two-stage spectral projection therefore provides (i) enhanced energy compaction, (ii) smoother cross-frequency regularity, and (iii) a representation that is both more structured and more learnable for the generative model.
>
> Lastly, related DWT–DCT–based spectral representations have been investigated in other medical imaging applications, though not in the context of fMRI BOLD signal modeling [1].
>
> **References**
> [1] Ghosh, S., Das, S., & Mallipeddi, R. (2021). A deep learning framework integrating the spectral and spatial features for image-assisted medical diagnostics. Ieee Access, 9, 163686-163696.

---

> > ### Author Response · Authors · 2025-12-04
> > **Minor Weakness 1,2,3,4:**
> >
> > # Minor Weaknesses
> >
> > **1/ In the abstract, "preserves key physiological brain dynamics" is an over-claims evidence.**
> >
> > **2/ There are numerous typos, for example: "we stitches the patches", "Fig. 1 provide an overview", etc**
> >
> > **3/ Some acronyms are undefined or defined too late, for example: MDD, STFT, etc. Even for some common acronyms, a brief reminder would improve readability**
> >
> > **4/ Please cite Kawahara et al., 2017 when mentioning BrainNetCNN so the provenance is clear.**
> >
> > # Response
> > We thank the reviewer for pointing out these minor but important issues regarding phrasing, acronym usage, and citation clarity.
> >
> > First, we agree that the original abstract wording preserves key physiological brain dynamics that may appear overstated. In the revision, we have incorporated a dedicated Neurophysiological Plausibility Analysis in section 4 that provides qualitative and quantitative evidence, including rsHRF and PSD alignment to support our statement.
> >
> > Second, we have carefully reviewed the manuscript and corrected typographical errors such as “we stitches the patches” and “Fig. 1 provide an overview.”
> >
> > Third, all acronyms (e.g., MDD, STFT) are now defined upon first use, and brief reminders are added where appropriate to improve readability. Finally, we have added explicit citations to Kawahara et al., 2017 wherever BrainNetCNN is mentioned to ensure the provenance of the method is clearly acknowledged. These revisions are incorporated in the updated manuscript.

---

> ### Author Response · Authors · 2025-12-04
> **Major Weakness 2,3,5**
>
> # Weakness:
> **2/ If B is too large, it will squash multiple ROIs into a single patch, which in turn will make it impossible for the flow to learn the interactions between those ROIs (since they are collapsed). This is not specifically discussed in the paper, and the experimental values for B are not provided.**
>
> **3/ In the contributions, you state that you "forming a unified dual-spectral image transform to capture both global and local spatiotemporal and spectral features for fMRI BOLD signal generation"; It is not clear what specifically differentiates your method from prior approaches in capturing ROI-to-ROI interactions (see 2/). Why should these interactions be better captured in spectral space rather that in the time domain? Note that classical fMRI analysis choose to capture the ROIs' interplays in the time domain.**
>
> **5/ Lines 60 to 63: It is not clear why your approach, which operates in the spectral domain, differs from methods that operate on preprocessed time series in its ability to filter out confounds; the preprocessing pipeline should have already remove cardiac pulsations, respiratory cycles, and motion-induced artifacts. Please clarify which components remain to be removed (see 4/).**
>
> # Response:
> We thank the reviewer for raising the important questions regarding the choice of block size B and why modeling ROI–ROI interactions in the spectral domain is advantageous compared to the raw time domain. First, we have added an ablation study in Table 5 evaluating different block sizes (B=2 and B=4) in the revised manuscript. The results show that the overall generation performance remains consistent across B, with a trade-off of a smaller B preserves finer frequency-time dependencies but leads to slower training due to the increased number of patches, whereas a larger B risks collapsing multiple ROIs-time into a single patch, thereby weakening local-temporal dependencies. Empirically, B=4 offers the best balance between preserving spectral-spatiotemporal locality and maintaining computational efficiency.
>
> Second, although standard preprocessing removes major physiological confounds (e.g., cardiac, respiratory, motion), the preprocessed BOLD signal remains inherently non-stationary, and its ROI–ROI interactions fluctuate over time due to dynamic connectivity. Modeling these interactions directly in the time domain forces the generative model to learn from transient correlation structures that are unstable and highly sensitive to short-lived fluctuations. In contrast, our dual-spectral representation via wavelet decomposition and DCT smoothing produces a localized time–frequency representation that stationrizes the signal. The wavelet transform isolates transient local dynamics, while the DCT compresses slow-varying spectral trends and suppresses residual high-frequency noise.
>
> This dual-stage spectral representation exposes the stable oscillatory components underlying in brain region coupling, making ROI–ROI interactions in the spectral domain more identifiable and substantially easier for the flow-matching model to learn than in the raw temporal domain.

---

> ### Author Response · Authors · 2025-12-04
> **Major Weakness 4**
>
> # Weakness
> **4/ Truncation is crucial since it removes part of the information (note that, at this stage, most artifacts should already have been filtered out by preprocessing); yet it is not specifically discussed in the paper, and the experimental values are not provided. Additionally, for short TRs (< 0.7 s), it remains unclear whether truncation would not be detrimental.**
>
> # Response:
>
> We thank the reviewer for highlighting the importance of truncation. Our framework includes an optional truncation step, we did not apply truncation in the experiments reported in the paper. All coefficients were retained to ensure no loss of information. The truncation module is included in our code as an optional tool to apply additional noise suppression on the data quality.
>
> Our proposed model operates in the time-frequency domain, making the repetition time (TR), or equivalently the sampling frequency an important hyperparameter to our framework. Since TR determines the maximum resolvable frequency according to the Nyquist theorem, it directly influences the spectral content of fMRI signals. Our framework can be tuned to match the frequency characteristics associated with a given TR, allowing compatibility with varying fMRI acquisition protocols. In contrast, time-domain diffusion models typically overlook TR and its impact on the spectral properties essential for accurately modeling fMRI dynamics.

---

> ### Author Response · Authors · 2025-12-04
> **Major Weakness 6**
>
> # Weakness
>
> **6/ The main issue with this paper is the lack of an exhaustive experimental validation. It should be evaluated on at least half a dozen datasets, spanning different TRs, spatial resolutions, and tasks (e.g., multiclass subject prediction).**
>
> # Response
>
> We thank the reviewer for highlighting this important point. Our experiments primarily use the Netsim dataset, a widely adopted benchmark from the FMRIB Analysis Group for evaluating unconditional time-series generation. To further demonstrate generalization, we also include experiments on a real-world Major Depressive Disorder (MDD) dataset, which evaluates class-conditional generation. We fully agree that evaluating publicly available datasets is valuable. Therefore, we additionally incorporate the ABIDE dataset, using a different parcellation scheme to assess the robustness of our model across heterogeneous datasets and anatomical parcellations. Specifically, we employ stratified 5-fold cross-validation for the MDD dataset with AAL 116 Atlases and 10-fold cross-validation for the ABIDE dataset with schaffer parcellation, providing extensive robustness assessments. The details of the dataset and the results are added in section 3.1 in our revised manuscript. We believe that current experiments on a total of 1 simulated and 2 real-world large Major Depression Disorder and Autism Brain Imaging Data Exchange fMRI dataset is sufficient to show the methodological advantages and application of our method. Further experiments on other brain disease dataset to evaluate generalizability will be reserved for future work.

---

> ### Author Response · Authors · 2025-12-04
> **Minor Weakness  5 and Question 5**
>
> # Minor Weakness  5 and Question 5
> **M5/ The results are somewhat ambiguous: the proposed approach is not SOTA in terms of "generation quality" (Table 1), yet it achieves SOTA prediction performance (Table 3). This discrepancy is surprising and is not discussed.**
>
> **Q5/ Do you have an explanation for the discrepancy between Table 1 and Table 3 (non-SOTA generation quality vs. SOTA prediction performance)?**
>
> # Response:
>
> We thank the reviewer for highlighting the discrepancy between the unconditional generation results in Table 1 and the downstream prediction performance in Table 3. We clarify that Table 1 evaluates unconditional generation quality on the NetSim dataset, which is a simulation benchmark in unconditional setting and does not reflect the class-conditional dynamics present in real-world clinical datasets. In contrast, Table 3 reports conditional generation results on the MDD dataset, where the model is guided by class labels.
> To avoid confusion, we have added the generation quality metrics for the MDD dataset in the revised manuscript (now incorporated into Table 3), presenting both generation and classification results. These results demonstrate that while unconditional NetSim generation is not SOTA, our model consistently achieves SOTA performance in conditional generation and downstream prediction on real data settings. This aligns with our primary goal of improving label-conditioned fMRI synthesis that is more aligned to real-world clinical applications.

---

> ### Author Response · Authors · 2025-12-04
> **Questions 3**
>
> # Questions 3
> **3/ Please report the computational cost of your approach: approximate training time, hardware used, and runtime for sample generation.**
>
> # Response
> We thank the reviewer for requesting computational cost. In the revised manuscript, we have added a dedicated section reporting the full runtime and resource usage of our approach. The training required 22 hours, 40 minutes, and 52.698 seconds of wall-clock time, while inference for generating the full samples took 48 minutes and 48.98 seconds with 1x A100 GPU. The model contains 130,844,352 parameters. These details are now included in the revision.

---

### Meta-Review · Area_Chair_GadG · 2026-01-17

**Summary:**

There is no consensus between reviewers.

Overall, I tend to agree with `YXrp`, that the manuscript has an innovative architecture and is well grounded theoretically. I also agree with `YXrp`, `7U7B`, `UQBR` that the experimental setting is limited. Even though the experimental design is sound, the breadth of experiments leaves much to be desired given the number of relatively large fMRI datasets (and the field's overall open data push).

However, I feel that the authors have responded partially to this, and in full to other criticisms.

**Reviewer Concerns:**

Even though the authors have responded, I feel that the "Limited physical and physiological interpretability" as described by `YXrp` remains as a problem. Even more so, `UQBR` raises general evaluation issues as main weaknesses. These are only partially closed as issues.

**Reviewer Scores:**

`YXrp` I suspect would increase their score.

---

### Decision · Program_Chairs · 2026-01-26

Accept (Poster)